# A novel RAB5 binding site in human VPS34-CII that is likely the primordial site in eukaryotic evolution

Saule Spokaite, Yohei Ohashi, Maxime Bourguet, Antoine Nicolas Dessus, Roger L Williams*

MRC Laboratory of Molecular Biology, Cambridge, United Kingdom

## eLife Assessment

This **convincing** study examines a novel interaction of RAB5 with VPS34 complex II. Structural data are combined with site-directed mutagenesis, sequence analysis, biochemistry, yeast mutant analysis, and prior data on RAB1-VPS34 and RAB5-VPS34 interactions to provide a new perspective on how RAB GTPases recruit related but distinct VPS34 complexes to different organelles. The judgment is that this work represents a **fundamental** advance in our understanding of VPS34 localization and regulation.

*For correspondence:
rlw@mrc-lmb.cam.ac.uk

Competing interest: The authors declare that no competing interests exist.

**Abstract** RAB5-GTP activation of the multiprotein VPS34 complex II (VPS34-CII) is critical for endosomal sorting and maturation, phagocytosis, and receptor downregulation. RAB5-GTP activates VPS34-CII by binding to a helical insertion in the C2 domain of VPS34 on the BECLIN1/UVRAG-containing adaptor arm of the complex. The autophagy complex, VPS34 complex I (VPS34-CI), features a unique ATG14L subunit in place of the VPS34-CII UVRAG subunit, and we found that this distorts the adaptor arm to alter the VPS34 RAB-GTPase binding pocket so that it preferentially binds RAB1-GTP. Surprisingly, our higher-resolution single-particle cryo-EM structure of VPS34-CII showed a second RAB5-GTP binding site on the VPS15 solenoid region. This site (VPS15-RAB5-site) appears to be the primordial RAB5-binding region. A mutant in the helical insertion of the C2 domain of human VPS34 that mimics the *Saccharomyces cerevisiae* sequence abolishes RAB5 binding to VPS34. Mutation of the VPS15-RAB5-site ortholog in *S. cerevisiae* VPS15 resulted in defective CPY sorting, loss of colocalisation with the RAB5 ortholog Vps21, and loss of binding to Vps21 in vitro. Evolutionary expansion from one to two RAB5-orthologue binding sites may have increased membrane binding and VPS34-CII activity to adapt to more complex endocytic systems.

## Introduction

Mammalian cells evolved three classes of phosphoinositide 3-kinases (PI3Ks). The class I and II enzymes are important elements of signal transduction pathways that evolved in metazoa (*Burke, 2018*; *Koch et al., 2021*; *Kücükdisli et al., 2023*; *Lo et al., 2023*; *Lo et al., 2022*; *Rathinaswamy and Burke, 2020*; *Vadas et al., 2011*; *Vogt et al., 2023*), but the class III PI3K, VPS34, is the primordial PI3K, which is present in all clades of eukaryotes. VPS34 catalyzes phosphate transfer from ATP to the D3-OH of phosphatidylinositol, producing phosphatidylinositol 3-phosphate (PI3P), which is important for cellular sorting pathways, including maturation of endosomes, phagocytosis, and autophagy. Also, recent studies suggest that VPS34/PI3P-dependent pathways can be hijacked by viruses (*Dahmane et al., 2022*; *Gong et al., 2024*; *Wang et al., 2021*; *Williams et al., 2021*; *Wu et al., 2024*). VPS34 is a component in two complexes, VPS34 complex I (VPS34-CI) and VPS34-complex II (VPS34-CII). The

complex-specific subunits cause VPS34-CI to localise to autophagosome and VPS34-CII to endosome membranes. In mammalian cells, VPS34 complexes have four core subunits. Both complexes share the VPS34, VPS15, and BECLIN1 subunits. In addition, VPS34-CI has ATG14L, while VPS34-CII has UVRAG. In yeast, the homologous complexes have common subunits known as Vps34, Vps15, and Vps30, while the unique subunits are Atg14 and Vps38 (*Itakura et al., 2008*; *Kihara et al., 2001*). VPS34-CI and VPS34-CII have been studied by X-ray crystallography and cryo-EM. These studies revealed that the overall structures of both complexes are similarly V- or Y-shaped, with one arm (the kinase arm) composed of most of the VPS34 and VPS15 subunits and the other arm (adaptor arm) consisting of BECLIN1 with ATG14L in VPS34-CI and BECLIN1 with UVRAG in VPS34-CII (*Baskaran et al., 2014*; *Cook et al., 2025*; *Ma et al., 2017*; *Rostislavleva et al., 2015*; *Tremel et al., 2021*; *Young et al., 2016*; *Young et al., 2019*).

We recently found that organelle-specific recruitment of VPS34 complexes is achieved by RAB GTPases: ER/autophagosome-localised RAB1A interacts with and activates VPS34-CI, while early endosome-localised RAB5A interacts with and activates VPS34-CII (*Tremel et al., 2021*). The 'Switch' regions in the RAB proteins undergo substantial conformational changes when switching from GDP- to GTP-bound states (*Pylypenko et al., 2018*). RAB effectors interact with these regions and selectively recognize RABs in a nucleotide-dependent manner (*Grosshans et al., 2006*). A triad of invariant hydrophobic residues F, W, and Y located in Switch 1, Interswitch, and Switch 2 regions, respectively, can adopt different conformations to specifically bind effectors (*Merithew et al., 2001*). While VPS34-CII will bind soluble RAB5 that is not coupled to membranes, this binding has no influence on the VPS34 activity (*Tremel et al., 2021*). To activate VPS34-CII, RAB5 must be coupled to membranes containing PI3P (*Tremel et al., 2021*). VPS34 complexes will phosphorylate soluble lipids, for example diC8PI, but their activities on membranes are greatly dependent on the charge and lipid saturation of the lipid membrane (*Ohashi et al., 2020*). Single-molecule kinetic studies are consistent with RAB5A activating VPS34-CII by both recruitment and allosteric activation (*Buckles et al., 2020*).

Our tomographic reconstruction of VPS34-CII on membranes provided a framework for understanding the mechanisms of VPS34-CII activation by RAB5A (*Tremel et al., 2021*). However, to improve the resolution of VPS34-CII, relative to the tomographic reconstruction, we have here determined high-resolution single-particle cryo-EM structures of VPS34-CII bound to RAB5A. Remarkably, we find that VPS34-CII-RAB5A complex has two independent RAB5A binding sites: one on VPS34 and one on the helical solenoid of VPS15. However, only the VPS34 site was detected in the previous tomographic reconstruction. This VPS34 site has three walls (a tripartite binding site) made of the VPS15 small globular domain (SGD, residues 723–805), the VPS15 WD40 (residues 968–1358), and the helical hairpin insertion in the VPS34 C2 domain (VPS34 C2HH, residues 170–210; *Cook et al., 2025*; *Tremel et al., 2021*). We previously attempted to eliminate RAB binding to VPS34 complexes by structure-guided mutation of the 199-REIE-202 stretch in VPS34 C2 HH helix 2 to AAAA (REIE >AAAA). VPS34-CI REIE >AAAA could neither bind nor be activated by RAB1A, while, surprisingly, the same mutation in VPS34-CII increased affinity for and activation by RAB5A (*Tremel et al., 2021*). The starkly different effects of this mutation on the two complexes remained unexplained by the cryo-ET study of VPS34-CII but can be explained by the new high-resolution single-particle cryo-EM structures.

## Results

### High-resolution structure of a mutant VPS34-CII bound to RAB5A

Using a confocal microscopy-based bead-binding assay, we showed that mCherry-tagged VPS34-CII bound specifically to RAB5A (*Figure 1A and B*; *Figure 1—figure supplement 1A–C*). Interaction of VPS34-CII with RAB1A was too weak to measure (*Figure 1—figure supplement 1D*). Consistent with our previous activity assays, the VPS34-CII REIE >AAAA mutation in the VPS34 C2 domain bound RAB5A about three times more tightly than the WT complex (*Figure 1B*; *Figure 1—figure supplement 1E*; *Tremel et al., 2021*). Because the VPS34-CII REIE >AAAA mutant bound more tightly to RAB5A than WT VPS34-CII (*Figure 1B*), we used it for our single-particle cryo-EM sample of the VPS34-CII-RAB5A assembly, then built and refined a structure at a 3.2 Å resolution (PDB: 9*RX5*; *Figure 1—figure supplement 2A and B*; *Figure 1—figure supplement 3*). The structure of the mutant bound to RAB5A is remarkably similar to VPS34-CII-BATS-RAB5A on membranes that we determined previously at low resolution by cryo-ET (PDB: 9S47, *Tremel et al., 2021*). There were only small changes in kinase

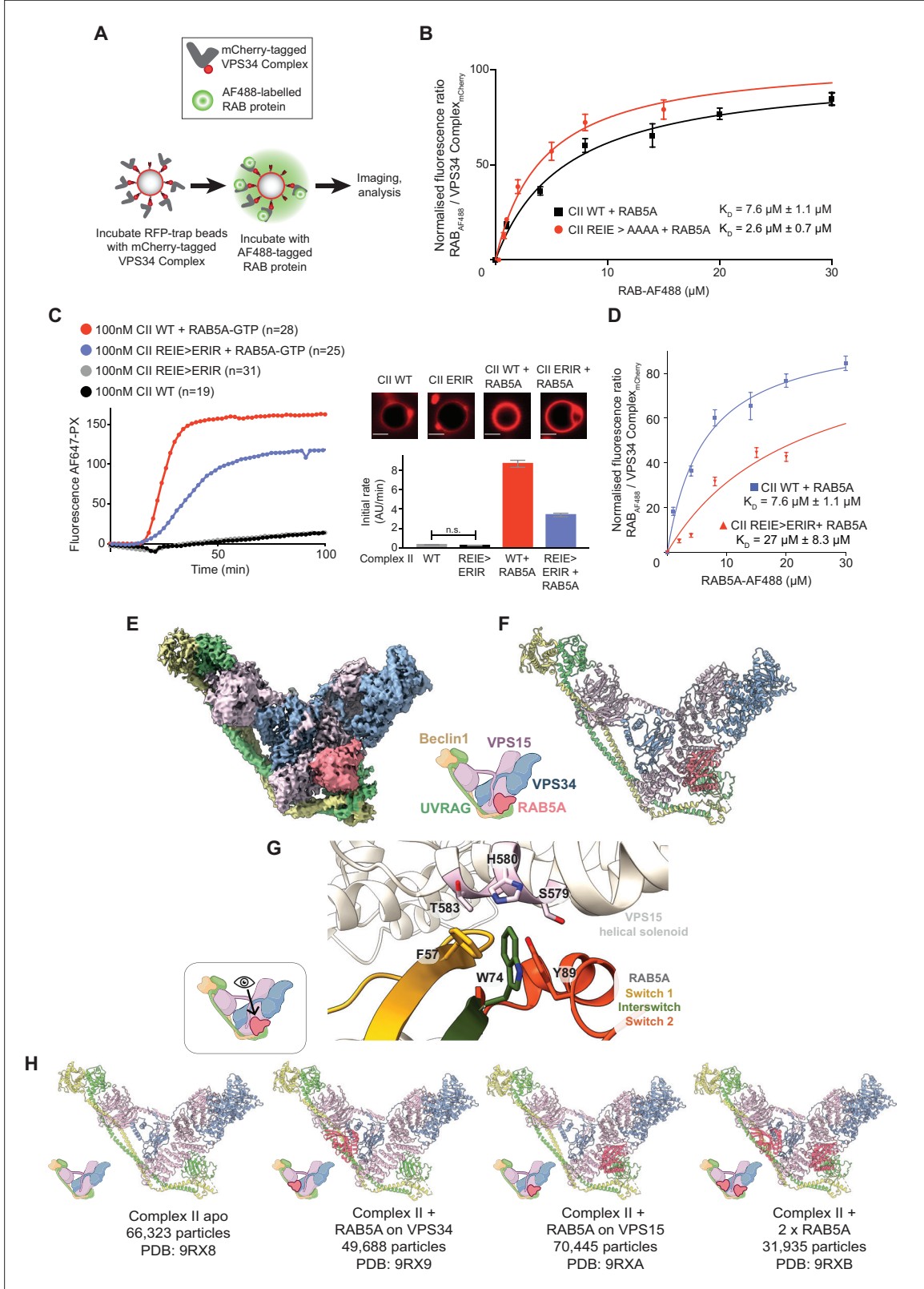

**Figure 1.** A Cryo-EM structure of human VPS34-CII with RAB5A showed a second RAB5 binding site on VPS15. Icons of VPS34-CII in E-H are based on an icon representing vps34-complex-ii created using BioRender.com. (**A**) Design of VPS34 CII – RAB5 binding affinity measurement assay. RFP-trap beads were coated with mCherry-tagged VPS34-CII, incubated with various concentrations of AF488-labelled RAB protein and imaged by confocal microscopy. (**B**) Quantification of RAB5-AF488 binding to VPS34-CII. Fluorescence intensity in the AF488 channel was normalised by the fluorescence

*Figure 1 continued on next page*

*Figure 1 continued*

intensity of the mCherry channel. Plots are representative of three independent replicates for each experiment, and the reported $K_D$ is average ± SD from three experiments. Error bars in the plots are mean ± SD of the representative experiment, with at least eight beads measured for each concentration (Micrographs are in *Figure 1—figure supplement 1C and E*). $K_D$ values are significantly different according to a t-test with a threshold of p<0.005. We verified that mCherry-tagging does not structurally compromise VPS34-CII (*Figure 1—figure supplement 1A*) and that mCherry alone does not interact with RAB proteins (*Figure 1—figure supplement 1B*). (**C–H**) VPS34-CII has two RAB5-binding sites, one on VPS34 and the other on VPS15. The VPS34-CII 199-REIE-202>ERIR reverse charge mutation in VPS34 subunit eliminates RAB5A binding to the VPS34 site, but not the VPS15 RAB5-binding site. (**C**) GUV-based assay with the wild-type VPS34-CII and VPS34-CII carrying an REIE >ERIR mutation in VPS34 C2 HH, in the presence and absence of membrane-tethered RAB5A. The REIE >ERIR mutation partially reduces VPS34-CII activation by RAB5A. Reaction progress curves are shown on the left, and initial rates on the right as bar graphs. All means are significantly different according to a one-way ANOVA test with p<0.05, except where not significantly different means are indicated by 'n.s.'. Experiments were performed in triplicate; error bars show ± SD of the representative experiment. Micrographs show the AF647-PX signal at the end of the reaction, scale bar = 5 µm. The time course with error bars is shown in *Figure 1—figure supplement 4*, error bars represent ± SD at each time point, with at least 19 GUVs measured. (**D**) The REIE >ERIR mutation reduces RAB5A binding affinity compared to the wild-type VPS34-CII. The plot shows binding curves of RAB5A and VPS34-CII REIE >ERIR mutant together with the wild-type VPS34-CII+RAB5A binding curves. Plots are representative of three independent replicates for each experiment. The reported binding $K_D$ is average ± SD from three experiments. Error bars in the plots are mean ± SD of the representative experiment, with at least six beads measured for each concentration (Micrographs are in *Figure 1—figure supplement 1C and F*). $K_D$ values are significantly different according to a t-test with a threshold of p<0.005. (**E**) Composite cryo-EM map of VPS34-CII REIE >ERIR with RAB5A bound to VPS15 subunit. (**F**) Cartoon representation of the atomic model for VPS34-CII REIE >ERIR-RAB5A. (**G**) A close-up view of the RAB5A binding site on VPS15. The RAB5A hydrophobic triad engages VPS15 helical solenoid domain. RAB5A residues F57 (yellow) and W74 (green) interact with VPS15 residue H580, RAB5A Y89 (orange) contacts VPS15 residues S579 and H580. (**H**) Atomic models of four different VPS34-CII states obtained from a cryo-EM sample of the WT VPS34-CII+RAB5A. Number of particles used for reconstruction of each state is given below the models. VPS34-CII was observed in apo state, with RAB5A bound to VPS34 C2 HH, with RAB5A bound to VPS15, or with RAB5A molecules bound to each site at the same time.

The online version of this article includes the following source data and figure supplement(s) for figure 1:

**Source data 1.** Raw and analysed data for GUV assays shown in *Figure 1C*, and bead assays shown in *Figure 1B and D*.

**Figure supplement 1.** RAB binding by wildtype and mutant VPS34 complexes.

**Figure supplement 1—source data 1.** NanoDSF data for *Figure 1—figure supplement 1A*.

**Figure supplement 1—source data 2.** Raw and analysed data for the bead assay shown in *Figure 1—figure supplement 1D*.

**Figure supplement 2.** Analysis of the cryo-EM structure of VPS34-CII with the VPS34 199-REIE-202>AAAA mutation bound to RAB5A.

**Figure supplement 3.** Single-particle cryo-EM data processing workflow for VPS34-CII REIE >AAAA-RAB5A (the 199-REIE-202>AAAA mutation is in the VPS34 subunit).

**Figure supplement 4.** GUV-based kinase assay reaction curves with error bars for results shown in *Figure 1C*.

**Figure supplement 5.** Single-particle cryo-EM data processing workflow for VPS34-CII REIE >ERIR-RAB5A (the 199-REIE-202>ERIR mutation is in the VPS34 subunit).

**Figure supplement 6.** HDX-MS shows protection in the two RAB5A binding regions in WT VPS34-CII.

**Figure supplement 6—source data 1.** Hydrogen-deuterium exchange mass spectrometry (HDX-MS) data for VPS34-CII vs VPS34-CII+RAB5A.

**Figure supplement 7.** Single-particle cryo-EM data processing workflow for WT VPS34-CII-RAB5A.

and adaptor arms of the complex (*Figure 1—figure supplement 2C*). The VPS15 helical solenoid was the best resolved region of the complex, while N- and C-terminal extremities of the BECLIN1 and UVRAG subunits were less well resolved. RAB5A bound to a tripartite site composed of VPS15 SGD, VPS15 WD40, and VPS34 C2HH (*Figure 1—figure supplement 2D and E*), which is consistent with our previous results (*Tremel et al., 2021*). We used this mutant complex as an initial model for building the structure of wild-type VPS34-CII-RAB5A (see below). One surprising feature of the mutant reconstruction was that there was weaker density that resembled a second molecule of RAB5A bound to the helical solenoid of VPS15 (*Figure 1—figure supplement 2A*). A model was built into this RAB5A density at partial occupancy, but this second site was even more apparent in the structure of a mutant in which the VPS34 RAB5A site was mutated (see next section).

## Structures of RAB5A-binding mutants suggest that VPS34-CII has an unexpected second RAB5A binding site on the VPS15 subunit

To examine the structural contribution of the VPS34 C2 HH to RAB5A binding, we introduced a reverse charge mutation in VPS34, 199-REIE-202–199-ERIR-202 (REIE >ERIR), to disrupt interactions between RAB5A Switch 1 and VPS34 C2 HH helix 2. Interestingly, the GUV kinase assay showed that although

the reverse charge mutation did not eliminate VPS34-CII activation by RAB5A, the extent of activation was reduced twofold compared to wild-type VPS34-CII (*Figure 1C*; *Figure 1—figure supplement 4*). Compared to wild-type VPS34-CII, the reverse charge REIE >ERIR mutant VPS34-CII had a RAB5A binding affinity that was drastically reduced, to a $K_D$ of 27±8.3 μM (*Figure 1D*; *Figure 1—figure supplement 1F*). To investigate whether the partial reduction in VPS34-CII REIE >ERIR activation by RAB5A was due to incomplete disruption of the VPS34-RAB5A interaction, we determined the structure of a VPS34-CII reverse charge mutant in the presence of RAB5A (*Figure 1E*; *Figure 1—figure supplement 5*). As expected, we did not observe any density in the tripartite RAB binding site. However, consistent with the weak density on the VPS15 subunit in our structure of VPS34-CII with the VPS34 REIE >AAAA mutation, we now saw strong density for RAB5A bound to the VPS15 subunit of VPS34-CII REIE >ERIR. We fit and refined an atomic model for RAB5A-GTP into the new density (PDB: 9RX6; *Figure 1F*) on the VPS15 helical solenoid domain, with an interaction centred on the helix spanning VPS15 residues 570–585. RAB5A mediates this interaction mostly through its hydrophobic triad residues F57, W74, and Y89 (*Figure 1G*). The Interswitch residue W74 contacts VPS15 residues S579 and H580, as well as the Switch 1 residue F57. In turn, F57 also interacts with VPS15 H580, and its nearby residue T583. The Switch 2 residue Y89 is not in direct contact with any VPS15 residues and instead binds to other RAB5A residues. This contrasts with the RAB5A binding to VPS34 C2 HH, where Switch 1 F57 has a stabilising role for the switch, but does not make direct interaction with the effector-binding site (*Figure 1—figure supplement 2E*). It is evident that RAB5A has generous plasticity in its hydrophobic triad region that allows binding to different molecular surfaces.

To verify that RAB5A binding the VPS15 site was not an artefact of cryo-EM sample preparation, we used hydrogen–deuterium exchange mass spectrometry (HDX-MS) and examined RAB5A binding to the wild-type VPS34-CII in solution (*Figure 1—figure supplement 6A*). We observed strong protection upon RAB5A binding not only in the VPS34 C2 HH helix 2 (*Figure 1—figure supplement 6B*), but also in the VPS15 helical solenoid domain, including helix 570–585 (*Figure 1—figure supplement 6C*). Protection in the VPS34 RAB5A binding site is consistent with our previous HDX-MS results (*Tremel et al., 2021*), and protection on the VPS15 helical solenoid residues 570–585 indicates that RAB5A binds the VPS15 site both in solution and in the cryo-EM sample.

## Wild-type VPS34-CII can bind two copies of RAB5A simultaneously

We next determined the structure of wild-type VPS34-CII-RAB5A to further confirm the presence of the new RAB5A binding site on VPS15 (*Figure 1—figure supplement 7*). After several rounds of heterogeneous refinement and 3D classification, we found that 30% of particles were apo VPS34-CII, 23% of particles had RAB5A bound to VPS34 only (VPS34-CII-RAB5A$_{VPS34}$), 32% of particles had RAB5A bound to VPS15-only (VPS34-CII-RAB5A$_{VPS15}$), and 15% of particles had RAB5A bound to both VPS34 and VPS15 (VPS34-CII-(RAB5A)$_2$; *Figure 1H*). We subjected these particles to numerous rounds of focused 3D classifications to ensure that the density of VPS34-CII-(RAB5A)$_2$ originated from particles having two bound RAB5As and was not an artefact of combining VPS34-CII-RAB5A$_{VPS15}$ particles with VPS34-CII-RAB5A$_{VPS34}$ particles. Apart from the binding of two RAB5As, the structure of VPS34-CII-(RAB5A)$_2$ did not show significant conformational differences from the other RAB5-bound VPS34-CII structures, nor from the apo form. This indicates that VPS34-CII can interact with two RAB5A GTPases simultaneously, via two independent binding sites. The structure of the VPS34 REIE >AAAA mutant that bound RAB5 tighter than the WT VPS34 also has a second RAB5 binding site on VPS15; however, it is likely that the population of complexes in the cryo-EM sample contained both particles with and without RAB5A bound to VPS15, since the average density in the VPS15 RAB5 binding site was only about 40% of the average density in the VPS34 RAB5 binding site. Models were built and refined for all RAB5-associated VPS34-CII assemblies.

To test whether the RAB5A binding site on VPS15 has a role on VPS34-CII activation, we mutated the VPS15 579-SHMIT-583 stretch into 579-DDMIE-583, with the expectation that the charged, bulkier aspartic and glutamic acid residues would disrupt the RAB5A hydrophobic triad interaction with VPS15. As expected, the mCherry-tagged VPS34-CII VPS15 mutant could be activated by RAB5A, but about twofold less than the wild-type VPS34-CII (*Figure 2A*; *Figure 2—figure supplement 1*). The VPS34-CII VPS15 mutant had a $K_D$ of 6.0±0.1 μM for RAB5A (*Figure 2B*; *Figure 2—figure supplement 2*), which is comparable to wild-type VPS34-CII, where $K_D$ = 7.4 ± 1.1 μM. To completely abolish RAB5A binding and activation of VPS34-CII, we combined the VPS34 REIE >ERIR and VPS15

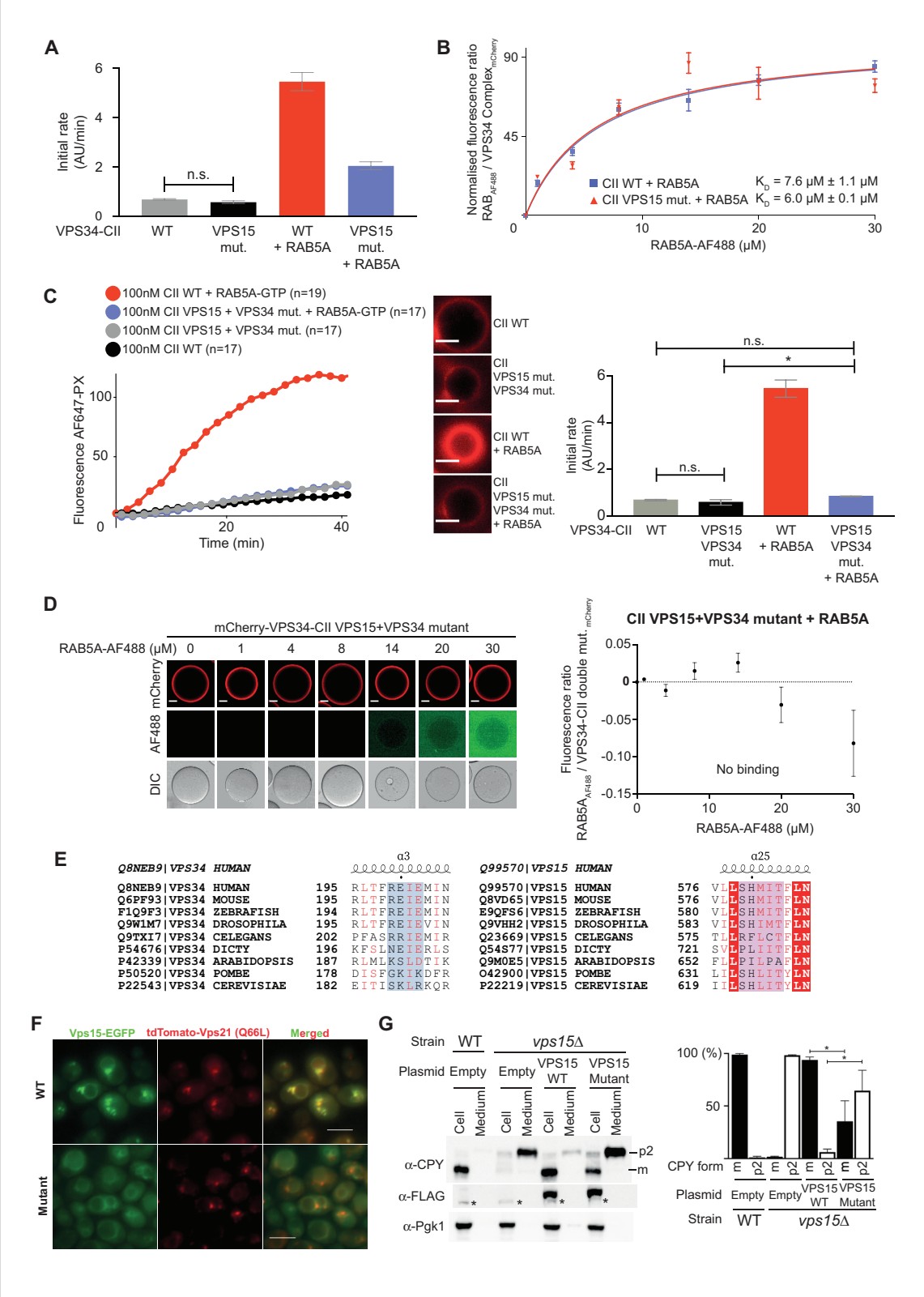

**Figure 2.** The VPS15 RAB5 binding site of VPS34-CII is necessary for full activity and is likely the primordial RAB5-binding site. (**A**) The human VPS15 579-SHMIT-583>DDMIE mutation partially reduced VPS34-CII activation by RAB5A. Bar graphs show initial rates from the GUV-based assay with the wild-type VPS34-CII and VPS34-CII carrying a 579-SHMIT-583>DDMIE mutation in the VPS15 helical solenoid (VPS15 mutant), in the presence and absence of membrane-tethered RAB5A. All means are significantly different according to a one-way ANOVA test with p<0.05, except where non significantly

*Figure 2 continued on next page*

*Figure 2 continued*

different means are indicated by 'n.s.'. Error bars show ± SD of the representative experiment. Experiments were performed in triplicate with mCherry-tagged VPS34-CII. Reaction curves for the assay are shown in *Figure 2—figure supplement 1*, where error bars represent ± SD at each time point, with at least 17 GUVs measured. (**B**) Binding curves for RAB5A interaction with VPS34-CII VPS15 mutant and with the wild-type VPS34-CII+RAB5. Plots are representative of three independent replicates for each experiment. The reported binding $K_D$ is average ± SD from three experiments. Error bars in the plots are mean ± SD of the representative experiment, with at least 6 beads measured for each concentration. $K_D$ values are significantly different according to a t-test with a threshold of p<0.005. Micrographs are in *Figure 2—figure supplement 2*. (**C**) The double mutation eliminates VPS34-CII activation by RAB5A, as shown by a GUV-based assay with the WT VPS34-CII and VPS34-CII carrying a double mutation (VPS34 C2 HH REIE >ERIR + VPS15 SHMIT >DDMIE). Reaction progress curves are shown on the left, and initial rates on the right as bar graphs. All means are significantly different according to a one-way ANOVA test with p<0.05, except where non significantly different means are indicated by 'n.s.'. Error bars are ± SD of the representative experiment. Micrographs show the AF647-PX signal at the end of reaction, scale bar = 4 µm. Experiments were performed in triplicate with mCherry-tagged VPS34-CII. The representative time course with error bars is shown in *Figure 2—figure supplement 4*, error bars represent ± SD at each time point, with at least 17 GUVs measured. (**D**) RAB5A does not bind VPS34-CII double mutant. Left: micrographs of RFP-trap beads coated with mCherry-VPS34-CII double mutant and incubated with increasing concentrations of RAB5A-AF488. Scale bar: 20 µm. Right: binding curve for RAB5A-AF488 binding to mCherry-VPS34-CII double mutant measured from images shown on the left. The plot on the right is representative of three independent replicates. Error bars in the plot are mean ± SD of the representative experiment, with at least six beads measured for each concentration. (**E**) The RAB5A binding site in VPS15, but not in VPS34 is conserved from humans to yeast. Left: multiple sequence alignment of VPS34 RAB binding site sequences. Right: multiple sequence alignment of VPS15 RAB5A binding site sequences. Identical residues are shown as white letters in red background. Red letters denote residues with similar properties. Structural elements shown above are based on VPS34 complex models from this study. (**F**) Yeast Vps15 carrying a 622-SHLITY-627>DDLIEY mutation in the Vps15 RAB5 binding site does not show strong colocalisation with a GTP-locked form of the yeast RAB5A orthologue, Vps21 (Q66L). C-terminally EGFP-tagged Vps15 (WT or mutant) was transformed into a vps15Δ strain integrated with tandem-tomato-tagged Vps21 (Q66L) and subjected to light microscopy. Scale bars: 5 µm. (**G**) Vps15 carrying a SHLITY >DDLIEY mutation in the RAB5 binding site is defective in CPY sorting. C-terminally 3 x FLAG tagged Vps15 (WT or mutant) was transformed into vps15Δ cells, and cellular and medium fractions ('cell' and 'medium', respectively) are subjected to western blotting. Left: Western blotting for CPY, FLAG for Vps15, and Pgk1. *: non-specific cross-react. Right: Quantification of mature (**m**) and precursor (**p2**) forms from triplicated experiments. Error bars in the plots are mean ± SD from three independent experiments. Significance was determined using a t-test with p<0.05 as significantly different (denoted as *).

The online version of this article includes the following source data and figure supplement(s) for figure 2:

**Source data 1.** Raw and analysed data for GUV assays shown in *Figure 2A and C*, and bead assays shown in *Figure 2B and D*; Western blot quantification for *Figure 2G*.

**Source data 2.** Raw, uncropped, unannotated western blot scans for *Figure 2G*.

**Source data 3.** Raw, uncropped western Blot scans for *Figure 2G* with relevant bands annotated.

**Figure supplement 1.** GUV-based kinase assay reaction curves with error bars for results shown in *Figure 2A*.

**Figure supplement 2.** RAB5A binding to VPS34-CII carrying a 579-SHMIT-583>579-DDMIE-583 mutation in VPS15.

**Figure supplement 3.** nanoDSF experiments show that VPS34-CII mutants that disrupt RAB5 binding are structurally intact.

**Figure supplement 3—source data 1.** NanoDSF data for *Figure 2—figure supplement 3A and B*.

**Figure supplement 4.** GUV-based kinase assay reaction curves with error bars for results shown in *Figure 2C*.

**Figure supplement 5.** The RAB5 binding mutation in Vps15 in yeast VPS34-CII abolishes binding to Vps21, a yeast RAB5 orthologue.

**Figure supplement 5—source data 1.** Raw data for the bead assays shown in *Figure 2—figure supplement 5*.

SHMIT >DDMIE mutations. We verified that all the single and double mutants were structurally intact by nanoDSF (*Figure 2—figure supplement 3*). The VPS15 +VPS34 double mutant VPS34-CII could not be activated by RAB5A (*Figure 2C*; *Figure 2—figure supplement 4*), and we observed no binding of RAB5A to the double mutant complex II – coated beads (*Figure 2D*). When the RAB5 binding sites in VPS34 and VPS15 are aligned with various species, the VPS34 RAB5-binding sequence REIE is not conserved in yeast *S. cerevisiae* and *S. pombe*, where it is rather reverse-charged (*Figure 2E*). In contrast, the VPS15 RAB5-binding sequences are highly conserved through evolution. Indeed, the yeast Vps15 RAB5 non-binding mutant (*S. cerevisiae* 622-SHLITY-627>DDLIEY) in yeast VPS34-CII was sufficient to abolish the binding to the yeast RAB5 orthologue, Vps21 in vitro (*Figure 2—figure supplement 5*). Also, the yeast Vps15 SHLITY >DDLIEY mutant did not show strong colocalisation with Vps21 (*Figure 2F*), and the same mutant is defective in endocytic sorting measured by CPY processing (*Figure 2G*). Taken together, these results show that RAB5A binds to two sites on human VPS34-CII, both sites are required for full activation, and that two RAB5A GTPases can bind VPS34-CII at the same time. It is likely that RAB5 binding in eukaryotic VPS34-CII originated at the VPS15 site, with the second binding site on VPS34 having been acquired during evolution.

## RAB5A binding to VPS34 site does not induce conformational changes in VPS34-CII

As described above, the cryo-EM sample of wild-type VPS34-CII-RAB5A contained both the apo and RAB5A-bound forms of VPS34-CII, which enabled us to confidently examine whether binding of RAB5 affected the conformation of the VPS34-CII. We could see that similarly to what was reported for RAB1A binding to VPS34-CI (*Cook et al., 2025*), RAB5A binding to VPS34-CII did not induce large conformational differences (*Figure 3A*; *Figure 3—figure supplement 1A–C*). Furthermore, the WT VPS34-CII-RAB5A structure did not differ greatly from VPS34-CII REIE >AAAA-RAB5A, nor a reinterpreted VPS34-CII-BATS-RAB5A model based on our cryo-ET data previously reported (*Tremel et al., 2021*; *Figure 3—figure supplement 1D and E*).

## RAB specificity for VPS34 complexes arises from RAB switch regions and from the VPS34 C2 HH conformation

We compared a previously reported VPS34-CI bound to RAB1A (VPS34-CI-RAB1A; *Cook et al., 2025*) with the VPS34-CII-RAB5A structure (*Figure 3*). Both RABs bind VPS34 complexes in the same tripartite site with similar orientations (*Figure 3A*). The most prominent element of the RAB binding site is the VPS34 C2 HH. In VPS34-CI, both helices of the C2 HH engage RAB1A (*Figure 3B*, left), while in VPS34-CII, only one VPS34 C2 HH helix engages RAB5A (*Figure 3B*, right). Also, in VPS34-CI, the VPS15 WD40 domain and RAB1A are separated by 27 Å (*Figure 3B*, left), while in VPS34-CII the distance between equivalent points in RAB5A and WD40 is only 9 Å (*Figure 3B*, right). These differences in the tripartite binding pocket are not due to RAB binding, since WT apo VPS34-CII has the same overall conformation as the RAB5-bound conformation (*Figure 3—figure supplement 1C*). Instead, the difference in the binding site appears to arise from the influence of the UVRAG subunit in VPS34-CII instead of the ATG14L subunit of the VPS34-CI. The more open RAB pocket adjacent to the VPS15 WD40 in VPS34-CI suggests that the WD40 might make a larger contribution to RAB binding in complex VPS34-CII than in complex VPS34-CI, indicating that the difference in the geometry of the tripartite site in the two complexes is the most important contribution to the RAB specificity.

In the VPS34 RAB-binding site, there are considerable differences between VPS34-CI and VPS34-CII in the interaction between the VPS34 C2 HH and the RAB hydrophobic triad, which is made of invariant hydrophobic residues F, W, and Y located in the Switch 1, Interswitch, and Switch 2 regions (*Figure 3C*). While the same elements are involved in both RAB-VPS34 complexes, they are arranged differently. In RAB1A (*Figure 3C*, left), Switch 1 residue F48 forms a cation-π stacking interaction with residue R195 in VPS34 C2 HH helix 2. Residue Y80 in Switch 2 interacts with R183 in VPS34 C2 HH helix 1. The Interswitch residue W65 does not engage VPS34 C2 HH, but instead contacts RAB1A residue F48 and others (*Figure 3—figure supplement 2A*). In contrast, for the VPS34-CII-RAB5A structure (*Figure 3C*, right), Interswitch residue W74 forms a cation-π stacking interaction with the VPS34 R195. In this complex, VPS34 R195 is also engaged by Y89 in the RAB5A Switch 2. RAB5A Switch 1 residue F57 does not directly participate in VPS34 binding and instead forms an intramolecular π-π stacking interaction with RAB5A W74.

In both RAB1A and RAB5A, the Switch 1 region is close to VPS34 C2 HH helix 2 (*Figure 3D*). In VPS34-CI-RAB1A, Switch 1 residue D47 forms a salt bridge with VPS34 R199, which, in turn, interacts with VPS34 E202 (*Figure 3D*, left). The importance of the VPS34 R199 - RAB1A D47 interaction is supported by our previous finding that the VPS34 REIE >AAAA mutation abolishes VPS34-CI recruitment and activation by RAB1A (*Tremel et al., 2021*). In VPS34-CII-RAB5A, Switch 1 of RAB5A is also positioned close to helix 2 of the VPS34 C2 HH (*Figure 3D*, right). However, instead of potential side chain hydrogen bonds or salt links between these elements, RAB5A has a non-polar A56 residue in the position equivalent to RAB1A D47 (*Figure 3—figure supplement 3*), and it is bordered by aliphatic residues I53, G54, and A55. This aliphatic RAB5A Switch 1 region interacts more favourably with the non-polar VPS34 AAAA mutant than with the polar REIE motif in WT VPS34 C2 HH helix 2 (*Figure 1—figure supplement 2E*), resulting in increased activation of VPS34-CII REIE >AAAA by RAB5A.

The Switch 2 region in both RABs interacts with different VPS34 C2 HH helices (*Figure 3E*). When bound to VPS34-CI, the flexible RAB1A Switch 2 loops around the positively charged R183 in VPS34 C2 HH helix 1, forming polar contacts with it via residue S78 and the backbone carbonyl oxygens of A68 and F73 (*Figure 3E*, left), with distances ranging from 2.8 Å to 3.4 Å, suggesting potential hydrogen bonding (*Figure 3—figure supplement 2B*). This conformation of Switch 2 contrasts with

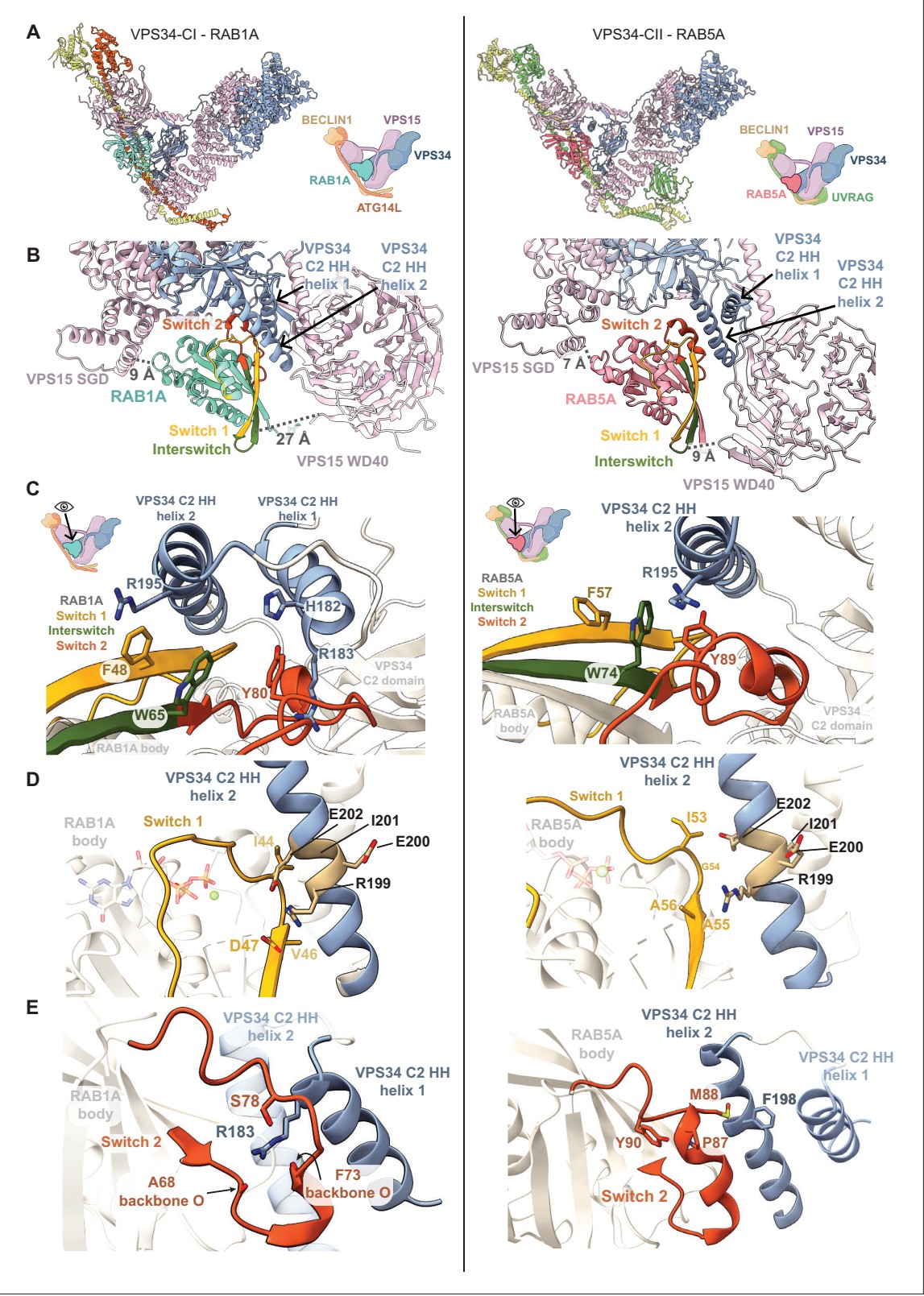

**Figure 3.** Comparison of VPS34-CI-RAB1A and VPS34-CII-RAB5A binding modes. Left panels show features of VPS34-CI-RAB1A (PDB: 9MHG). Right panels show features of VPS34-CII-RAB5A (PDB: 9RX9). Icons of VPS34 complexes are based on an icon representing vps34-complex-ii created using BioRender.com. (**A**) RAB1A and RAB5A bind to the equivalent site in both complexes. (**B**) RAB1A binds both helices of the VPS34 C2 HH. Switch 1 region (yellow) binds helix 2, while Switch 2 (orange) binds helix 1. RAB5A contacts only helix 2 from VPS34 C2 HH. The 9 Å distance between RAB1A

*Figure 3 continued on next page*

*Figure 3 continued*

and VPS15 SGD is comparable to the 7 Å distance between RAB5A and VPS15 SGD. The VPS15 WD40-RAB distance is larger in VPS34-CI-RAB1A (27 Å) than in VPS34-CII-RAB5A (9 Å). Distances were measured between the α-carbon atoms in selected VPS15 and RAB residues: VPS15 S776 and RAB1A T130/RAB5A N139 for RAB-SGD distance; VPS15 G1298 and RAB1A T59/RAB5A T68 for RAB-WD40 distance. The RAB residues are equivalent according to a multiple sequence alignment (*Figure 3—figure supplement 3*). (**C**) Hydrophobic triad residues in RAB1A and RAB5A bind VPS34 C2 HH in different modes. In RAB1A, residue F (yellow) contacts C2 HH helix 2, residue Y (orange) engages C2 HH helix 1, and residue W (green) is not involved in direct contact with the C2 HH. By contrast, in RAB5A, it is residue F (yellow) that is not involved in binding C2 HH, while both residues W (green) and Y (orange) engage the same VPS34 C2 HH helix 2. VPS34 C2 HH helix 2 residue R195 interacts with hydrophobic triad residues in both RAB1A and RAB5A. VPS34 residue R183 in helix 1 is involved in binding RAB1A, but not RAB5A. (**D**) Switch 1 region (yellow) interacts with VPS34 C2 HH helix 1 in both VPS34 complex-RAB assemblies. RAB1A residue D47 forms a salt bridge with VPS34 residue R199. RAB5A Switch 1 has limited contact with VPS34 C2 HH helix 1. (**E**) Switch 2 region (orange) adopts different conformations in VPS34-CI-RAB1A and VPS34-CII-RAB5A. In RAB1A, Switch 2 is in a loop conformation, embracing VPS34 residue R183 in VPS34 C2HH helix 1 that in return stabilises the loop via polar contacts to RAB1A S78 and the backbone carbonyls of A68 and F73. In RAB5A, Switch 2 is in a helical conformation potentially stabilised by a hydrophobic interaction between RAB5A residues Y90 and P87. Switch 2 makes minimal contact with C2HH helix 2 through residue M88 that interacts with F198 in VPS34.

The online version of this article includes the following figure supplement(s) for figure 3:

**Figure supplement 1.** Human VPS34-CII with and without RAB5A has minimal conformational differences compared to other VPS34-CII structures.

**Figure supplement 2.** Analysis of the cryo-EM structure of VPS34-CI-RAB1A (PDB 9MHG).

**Figure supplement 3.** Multiple sequence alignment of selected human RAB GTPases and their isoforms.

a more helical conformation in other RAB1A-effector complexes (*Cheng et al., 2012*; *Dong et al., 2012*; *Mishra et al., 2013*). In a complex with VPS34-CII, RAB5A Switch 2 adopts a compact helical conformation stabilised by an intramolecular interaction between residues P87 and Y90. The M88 residue in this helical Switch 2 makes a non-polar interaction with F198 in VPS34 C2 HH helix 2 (*Figure 3E*, right). Interestingly, RAB1A S78 and RAB5A P87 share the same position in the multiple sequence alignment (*Figure 3—figure supplement 3*), suggesting that the function of this position is to stabilise a RAB-specific conformation of the Switch 2 region.

Overall, to bind VPS34-CII, RAB5A uses two interfaces (Switch 1 and the hydrophobic triad) and relies on the tripartite site in VPS34-CII. In contrast, RAB1A binding to VPS34-CI involves three interfaces (Switch 1, Switch 2, and the hydrophobic triad) that make polar and nonpolar interactions with both VPS34 C2 HH helices, but does not appear to contact other elements of the VPS34-CI tripartite site.

## VPS34-CII has a zinc finger at the N-terminus of BECLIN1

The most obvious characteristic of higher-resolution VPS34-CII that we observed was the presence of an additional RAB5-binding site on VPS15, near the base of the V-shaped complex. A second surprising feature of our higher-resolution structures of human VPS34-CII was additional features at the base of the V-shaped complex, including the N-terminal end of BECLIN1 that previously were not apparent. While there is no indication that these BECLIN1 features are related to RAB binding, they are likely to be important for the function of VPS34-CII.

The BECLIN1 N-terminal region has two highly conserved CXXC motifs (18-CXXC-21 and 137-CXXC-140) flanked by an intrinsically disordered region (IDR; *Mukhopadhyay et al., 2024*), which has hindered detailed studies of this region (*Chen et al., 2023*; *Rostislavleva et al., 2015*; *Young et al., 2019*). Our cryo-EM densities allowed us to dock and refine atomic models of the BECLIN1 N-terminus predicted by AlphaFold3 (*Abramson et al., 2024*). In VPS34-CII, there is a $Zn^{2+}$ coordinated by the two BECLIN1 CXXC motifs to form a zinc finger (*Figure 4—figure supplement 1*). In addition, the N-terminal C2 domain of UVRAG could be clearly built. The structures show that the C2 domain is sandwiched between the N-terminal zinc finger of BECLIN1 and the long helical hairpin in the middle of the VPS15 solenoid (residues 465–545; *Figure 4*). There was a density at the edge of strand β2 of the UVRAG C2 domain that appeared to augment the second β-sheet of the C2 domain. The AlphaFold3 model also had an augmented β-sheet and we have attributed this augmenting strand to BECLIN1 residues 78–83. This mostly ordered and relatively compact N-terminal region of BECLIN1 and UVRAG contrasts with the analogous region in VPS34-CI that was almost completely disordered (*Cook et al., 2025*). The N-terminal region of BECLIN1 also contains the BH3 motif of BECLIN1 (residues 105–130) to which Bcl-2 family proteins can bind to inhibit autophagy. It is not immediately clear

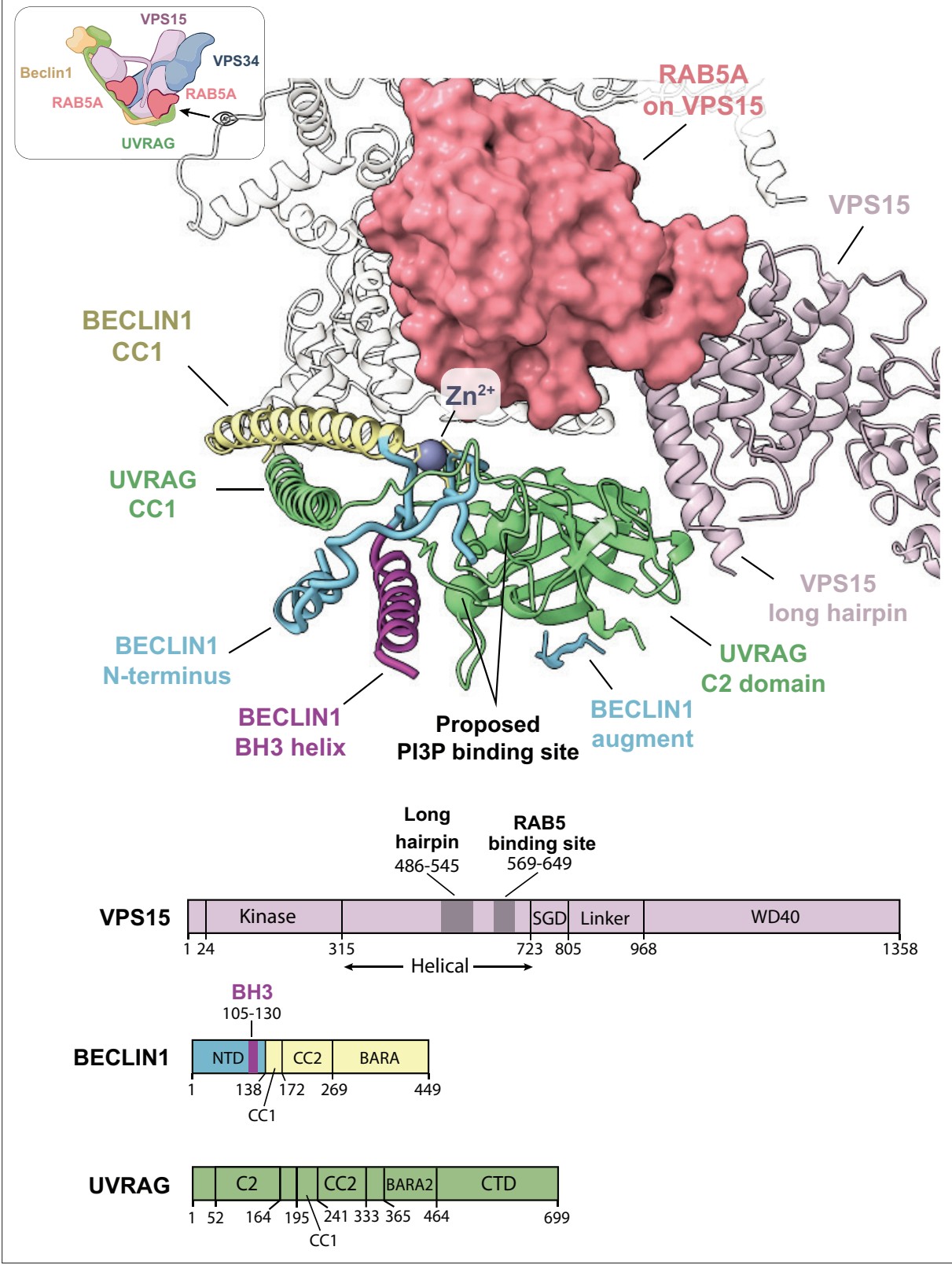

**Figure 4.** Structure of the N-terminal region of BECLIN1 and UVRAG (PDB: 9RXB). Icon of VPS34-CII is based on an icon representing vps34-complex-ii created using BioRender.com. Upper: The N-terminal region of BECLIN1 (blue) is stabilised by a zinc finger (Zn²⁺ grey), and this region together with the long helical hairpin in the VPS15 solenoid region sandwich the UVRAG C2 domain. Two basic residues (green spheres) comprise a proposed PI3P-

*Figure 4 continued on next page*

*Figure 4 continued*

binding region (*He et al., 2013*). RAB5 on VPS15 is indicated with an orange molecular surface. The BH3 helix of BECLIN1 is coloured magenta. Lower: linear domain compositions of VPS15, BECLIN1, and UVRAG. The colours correspond to equivalent domains in the cartoon.

The online version of this article includes the following figure supplement(s) for figure 4:

**Figure supplement 1.** VPS34-CII has a zinc finger at BECLIN1 N-terminus.

**Figure supplement 2.** Cryo-EM densities show that the N-terminus of VPS15 is myristoylated and VPS15 is bound to a GDP nucleotide.

how Bcl-2 binding would inhibit the VPS34 complexes, but the BH3 forms one of the two contacts between the UVRAG C2 domain and other subunits of the VPS34-CII (purple helix in *Figure 4*).

## Discussion

Our single-particle cryo-EM structures of assemblies of human VPS34-CII-RAB5A showed unanticipated aspects of VPS34 regulation. The human VPS34-CII VPS34 REIE >AAAA mutant bound RAB5A with a greater affinity than WT VPS34-CII and enabled the highest resolution VPS34-CII structure (3.2 Å). We have also determined apo structures of wild-type VPS34-CII at a resolution of 3.9 Å. These structures allowed us to delineate the molecular basis for RAB specificity of VPS34 complexes and determine that the differences in the overall shape of the VPS34 RAB binding pocket is not a consequence of RAB binding but a consequence of the complex-specific subunit (UVRAG versus ATG14L). We previously observed that VPS34-CII was not activated by RAB1 but that VPS34-CI is weakly activated by RAB5 in vitro (*Tremel et al., 2021*). This differential selectivity may arise from the tripartite RAB5 binding site in VPS34-CII being narrower than the equivalent site in VPS34-CI. This may restrict the geometry of RAB binding in a way that would only allow RAB5A, but not RAB1A, to fit and bind the tripartite site of VPS34-CII.

One subunit difference between the two apo complexes causes conformational changes in the VPS34 tripartite RAB-binding site. Most importantly, in VPS34-CI, the C2 HH is positioned with both helices available for RAB1A, while in VPS34-CII only one helix from the C2 HH is accessible for RAB5A binding. The diminished interaction with the C2 HH in the VPS34-CII-RAB5A assembly appears to be compensated by a narrowing of the tripartite RAB5A pocket with additional, non-switch interactions between RAB5A and VPS15. Remarkably, the complex-specific ATG14L and UVRAG subunits do not come in direct contact with RABs. Given that the apo-VPS34-CII closely resembles the VPS34-CII-RAB5A, it appears that ATG14L tunes binding to RAB1A by twisting the common subunits VPS34, VPS15 and BECLIN1 into a RAB1-specific conformation.

RAB1A and RAB5A bind VPS34-CI and -CII via the Switch and Interswitch regions. In both RAB-VPS34 complex assemblies, the RAB hydrophobic triad is positioned at the linker between the VPS34 C2 HH helices and engages it in a similar manner. In contrast, the rest of the RAB Switch regions make VPS34 complex-specific interactions. RAB1A Switch 1 residue D47 particularly stands out, as it forms a salt bridge with VPS34, while in RAB5A Switch 1 is mostly hydrophobic and forms entirely different contacts with VPS34. The hydrophobic triad likely provides most of the binding energy, and the Switch regions confer specificity, as has been observed with other RAB-effector complexes (*Lucas et al., 2014*). The distinctive, hydrophobic RAB5A Switch 1 region interaction with the VPS34 C2 HH also explained why the VPS34-CII RAB5A binding site mutation REIE >AAAA, which was meant to inhibit RAB5A binding, increased the binding affinity and activation of VPS34-CII (*Tremel et al., 2021*).

Our VPS34 complex structures showed an N-terminal VPS15 myristoylation and a GDP bound to the VPS15 kinase domain (*Figure 4—figure supplement 2*). The myristoyl-binding pocket is in the vicinity of the interface with VPS34 (*Figure 4—figure supplement 2*). It was proposed that VPS34 is activated when the myristoyl moiety is ejected from the VPS15 N-lobe and inserts into a lipid membrane, however, the same study also showed that a non-myristoylatable mutant (G2A) is fully functional (*Cook et al., 2025*), which is consistent with a previous yeast Vps15 study (*Herman et al., 1991*). Therefore, the precise role of the VPS15 myristoylation in activation remains to be elucidated. The myristoyl group inserting into membranes would increase association with membranes, but it also has been shown that RAB5 makes a very prominent contribution to the association with membranes (*Buckles et al., 2020*).

What (*Cook et al., 2025*) designated as the 'inactive' conformation (PDB ID 9MHG) resembles the conformation that we report here for our cryo-EM studies of VPS34-CII. In none of our datasets did we observe the minor 'active' conformation previously reported (*Cook et al., 2025*) which involved a 140° rotation of the VPS34 Helical-Catalytic (HELCAT) unit (PDB ID 9MHH). It may be that the VPS34-CII is poised for activation but not yet activated under the conditions in our cryo-EM study. It is also possible that the transition state on a membrane is sufficiently short-lived that we have not been able to capture it, even though we have RAB5 bound in the single-particle structures in the absence of membranes and RAB5 bound in the presence of membranes from our previous cryo-ET study (*Tremel et al., 2021*).

Surprisingly, we discovered a new RAB5A binding site on the VPS15 subunit of VPS34-CII. Like the previously known VPS34 site and other RAB-effector interactions (*Eathiraj et al., 2005*; *Hou et al., 2011*; *Lucas et al., 2014*), this newly discovered VPS15 site has micromolar-range affinity for VPS34-CII. In addition, the VPS15 RAB5 binding site is α-helical, which also is typical in RAB effectors (*Kawasaki et al., 2005*; *Rai et al., 2019*). The VPS15 site consists of a single α-helix while the VPS34 site has two helices.

Altogether the avidity of VPS34-CII for RAB5 is dictated by interactions in three disparate regions: helix 2 of the VPS34 helical insertion in the C2 domain, the VPS15 WD40 in the tripartite binding site, and the VPS15 solenoid interaction with separate RAB5. In contrast, VPS34-CI interacts with RAB1 using only a single region, the helical insertion in the VPS34 C2 domain. VPS34-CI appears to compensate for lack of the other two regions by interacting with both helices of the helical insertion in the VPS34 C2 domain. This is enabled by the unique bend of the ATG14L-containing adaptor arm that shifts the helical insertion to bring both helices into contact with bound RAB1.

In contrast to the VPS34 site, RAB5A binding to VPS15 does not involve extensive specific interactions through the Switch regions beyond the hydrophobic triad. This might account for the lower affinity of the VPS15 binding site compared to the VPS34 site. RAB1A does not bind or activate VPS34-CII, suggesting either that the conformation of the hydrophobic triad and the Switch regions in RAB1A are not complementary to VPS15 helix 570–585, or that the VPS15 RAB-binding site is blocked by other elements in VPS34-C1. RAB5 binding assays are consistent with a higher affinity VPS34 site and a lower affinity VPS15 site binding independently. However, functional assays indicate that each RAB5A binding site contributes to activation equally. The lack of synergy in the binding assays, despite clear synergy in activity assays is consistent with the complex binding to two RAB5A molecules on the same membrane. Such synergy could not be observed in the binding assays because of the way they were conducted: VPS34-CII was bound to beads and soluble RAB5A was titrated. To observe the synergy that we observe on membranes would require the RAB5 to be fixed to beads, while titrating VPS34-CII, and this arrangement would be impossible because of the limited solubility of the VPS34-CII. How the two RAB5A binding sites in VPS34-CII are coordinated in mammals remains to be seen.

There are only a few other cases where multivalent RAB effectors contain binding sites with different affinities (*Rai et al., 2019*). For example, MICAL1, a cytoskeleton regulatory protein, contains two RAB8 binding sites, one with high and one with low affinity for RAB8. The higher affinity site is used for initial MICAL1 recruitment to membranes, while the lower affinity site binds later (*Rai et al., 2019*). The two RAB5A binding sites on VPS34-CII might be a part of a similar mechanism that may control VPS34 activity on early endosomes. VPS34-CII would be fully activated only when present on highly concentrated RAB5A membrane domains and in absence of other RAB5A effectors that might bind RAB5A more tightly than VPS34-CII VPS15 site. On the other hand, VPS34-CII may cooperate with RAB5A effectors EEA1, Rabaptin-5, and Rabenosyn-5 to maintain early endosomal RAB5A domains that are required for endosome tethering and fusion (*Zerial and McBride, 2001*). In addition, VPS34-CII could also be involved in vesicle tethering if two RAB5 proteins from different vesicles bound to a single VPS34-CII. More sensitive experiments such as single molecule analysis will be required to investigate these possibilities. Finally, two RAB5A molecules might be required to correctly orient the VPS34 complex on the membrane, as both RAB binding sites are on the same side of the complex. While we did not observe two RAB5A proteins bound to VPS34-CII-BATS in the cryo-ET structure (*Tremel et al., 2021*), the lower occupancy of RAB5A on vesicles used for the cryo-ET structure on membranes might have meant that the VPS15 RAB5A binding site was not occupied.

Interestingly, multiple sequence alignments suggest that yeast Vps34-CII has a single RAB5 binding site, and this would be on the Vps15 subunit. As with human RAB5, the yeast RAB5 orthologue Vps21 is required in endocytic trafficking (*Singer-Krüger et al., 1994*), and the RAB1 homolog Ypt1 is involved in autophagy initiation (*Yao et al., 2024*). Furthermore, Rab-pulldown experiments identified direct or indirect interactions with Vps15 (*Graef et al., 2013*; *Nagano et al., 2024*). We showed that disrupting the Vps21 binding site in yeast Vps15 abolished the binding of Vps21 to yeast complex II in vitro, and inhibited endocytic sorting in vivo. It is possible that the RAB binding site on VPS15 evolved first, and the second RAB binding site on VPS34 evolved in higher organisms, as the complexity of membrane compartments increased the demand for tighter spatial and temporal regulation of PI3P production. It will be interesting to investigate how these RAB5A binding sites contribute to VPS34-CII function in human cells.

# Methods
## Lipid mixtures

| ID | Composition | Descriptor | Purpose |
|---|---|---|---|
| STGUV2 | 18% liver PI, 10% DOPS, 17% DOPE, 55% DOPC, 0.03% DO-Liss-Rhod-PE, 0.03% DSPE-PEG(2000)-Biotin | DO base | For assays without RABs |
| STGUV10 | 18% liver PI, 10% DOPS, 7% DOPE, 10% DOPE-MCC, 55% DOPC, 0.03% DO-Liss-Rhod-PE, 0.03% DSPE-PEG(2000)-Biotin | DO base +10% PE-MCC | For assays with RABs |

## Lipids

| Lipid | Manufacturer | Catalog number |
|---|---|---|
| Liver PI (mixed chain, Bovine) | Avanti Polar Lipids, Inc | 840042C |
| DOPS | Avanti Polar Lipids, Inc | 840035C |
| DOPE | Avanti Polar Lipids, Inc | 850725C |
| DOPC | Avanti Polar Lipids, Inc | 850375C |
| DO-Liss-Rhod-PE | Avanti Polar Lipids, Inc | 810150C |
| DSPE-PEG(2000)-Biotin | Avanti Polar Lipids, Inc | 880129C |
| DOPE-MCC | Avanti Polar Lipids, Inc | 780201C |

## Plasmids

| ID | Construct | Back-bone | Marker | Use | Reference | addgene # |
|---|---|---|---|---|---|---|
| pJB78 | GST-TEV-HsRAB1A Q70L C26S C126S,1–204 | pOPTG | amp | Cryo-EM, GUV assays | *Tremel et al., 2021* | 235738 |
| pYO1005 | HsUVRAG_FL, untagged | pCAG | amp | Bead assays | *Tremel et al., 2021* | 235702 |
| pYO1015 | HsVPS15_FL-3xTEV-ZZ | pCAG | amp | Cryo-EM, GUV assays | *Tremel et al., 2021* | 235704 |
| pYO1025 | HsVPS34_FL, untagged +hsVPS15-3xTEV-ZZ | pCAG | amp | Cryo-EM, GUV assays, Bead assays | *Ohashi et al., 2016* *Ohashi et al., 2020* *Tremel et al., 2021* *Buckles et al., 2020* | 235705 |
| pYO1031 | HsBeclin 1_FL +HsUVRAG_FL, both untagged | pCAG | amp | Cryo-EM, GUV assays | *Tremel et al., 2021* | 235706 |

*Continued on next page*

*Continued*

| ID | Construct | Back-bone | Marker | Use | Reference | addgene # |
|---|---|---|---|---|---|---|
| pYO1274 | HsVPS34 (REIE to AAAA)_FL, untagged | pcdna4/TO | amp | Cryo-EM, GUV assays | *Tremel et al., 2021* | 235709 |
| pYO1344 | HsVPS34 R199E E200R E202R_FL, untagged | pcdna4/TO | amp | Cryo-EM, GUV assays | This study | 235711 |
| pYO1280 | HsVPS34_FL WT-EGFP | pEEGFP-C1 | km | Cell biology | *Tremel et al., 2021* | 235712 |
| pYO1356 | HsVPS34 R199E E200R E202R_FL-EGFP | pcdna4/TO | amp | Cell biology | This study | 235713 |
| pYO1846 | mCherry-HsBeclin 1_FL | pCAG | amp | Bead assays | This study | 235714 |
| pYO1847 | HsVPS15 S579D_H580D_T583E_FL-3x tev-ZZ | pCAG | amp | Cryo-EM, GUV assays | This study | 235715 |
| pYO1865 | HsVPS15 S579D_H580D_T583E_FL, untagged | pCAG | amp | Cell biology | This study | 235716 |
| pYO350 | HsVPS15_FL, untagged | pcDNA4/TO | amp | Cell biology | *Tremel et al., 2021* | 235717 |
| STp8 (pOP823) | HsRAB5A Q79L C19S,C63S, 1–212 | pOPINS | kan | GUV assays, CryoEM | *Tremel et al., 2021* *Buckles et al., 2020* | 235718 |
| pYO1645 | GST-PX domain with Cys at 2 and C85S | pOPTG | amp | GUV assays | *Buckles et al., 2020* | 235719 |
| pYO634 | ScVPS15(promoter +WT FL coding)-EGFP | pRS416 | amp | Yeast experiments | This study | 235720 |
| pYO1880 | ScVPS15 (promoter +S622D_H623D_T626E_FL coding)-EGFP | pRS416 | amp | Yeast experiments | This study | 235721 |
| pYO950 | ScVPS15 (promoter +WT_FL coding)3xFLAG | pRS416 | amp | Yeast experiments | This study | 235722 |
| pYO1877 | Scvps15 (promoter +S622D_H623D_T626E_FL coding) –3xFLAG | pRS416 | amp | Yeast experiments | This study | 235723 |
| pYO1510 | mCherry-TEV-linker-14xhis | pOP | amp | Bead assays | This study | 235739 |
| pYO1006 | HsBECLIN1 untagged | pCAG | amp | GUV assays | *Tremel et al., 2021* *Ohashi et al., 2020* | 235724 |
| pYO1879 | PHO5 pro-dRFP-ScVPS21 Q66L-LEU2 | pBluescript (KS-) | amp | Yeast experiments | This study | 242480 |
| pYO1919 | ScVPS21 Q66L, 1–207 A-EGFP | pOPTH | amp | Bead assays | This study | 235769 |
| pRS416 | Control, backbone vector | pRS416 | amp | Yeast experiments | *Sikorski and Hieter, 1989* | GenBank: U03450.1 |
| pYO1920 | ScVPS34, untagged +ScVPS15 S622D_H623E_T626E-ZZ | pRS426 | amp | Bead assays | This study | 235767 |

## Protein purification

### Human VPS34-CII

To prepare VPS34-CII, Expi293 suspension cells (Thermo Fisher A14527) were grown in Expi293 Expression Medium (Thermo Fisher A1435102) at 37 °C, 8% $CO_2$, shaking at 125 rpm. At density of $2 \times 10^6$/L, cells were transfected with 1.1 mg DNA/L cells using polyethyleneimine 'MAX' (Polysciences 24765, 1 mg/mL in PBS) at 3 mg/L cells as the transfection reagent. Cells were grown further in the same conditions as above for 68 hr, then harvested by centrifugation at 3000 × *g* for 15 min, flash frozen in liquid nitrogen, and stored at –80 °C until use. The Expi293 cells were authenticated using short tandem repeat (STR) genotype analysis. They were not routinely tested for mycoplasma contamination.

To purify the protein, cells were resuspended in 50 mL/2 L cells of lysis buffer (50 mM HEPES pH 8.0, 300 mM NaCl, 1% Triton X-100 [Fisher BioReagents BP151], 12% glycerol, 2 mM $MgCl_2$, 1 mM

phenylmethylsulfonyl fluoride [PMSF, Thermo Fisher Scientific 36978], 0.5 mM tris(2-carboxyethyl) phosphine [TCEP], and 1 x EDTA-free protease inhibitor cocktail tablet [Roche 11873580001]) and incubated with slow stirring. Lysate was cleared by centrifugation at 142,000 × $g$ for 30 min at 4 °C in a Type 45 Ti rotor (Beckman Coulter) and filtered through a 5-µm syringe filter (Sartorius 17594). Cleared lysate was incubated for 3.5 hr at 4 °C and 8 rpm rotation with 3 mL IgG beads (Cytiva 17–0969) equilibrated with lysis buffer. Beads were then transferred to a gravity column, washed with 50 mL ice-cold wash buffer (50 mM HEPES pH 8.0, 300 mM NaCl, 0.1% Triton X-100, 12% glycerol, 50 mM MgCl$_2$, 0.5 mM TCEP, 4 µg/mL bovine pancreatic RNase [Sigma 83834]), and incubated with 50 mL wash buffer supplemented with 5 mM ATP for 20 min. Beads were further washed with another 50 mL wash buffer and 100 mL TEV buffer (50 mM HEPES pH 8.0, 300 mM NaCl, 0.5 mM TCEP), then resuspended in 10 mL TEV buffer and incubated with 40 µL of 4.4 mg/mL TEV protease (house made) overnight without rotation to remove the C-terminal ZZ-tag on VPS15. Flow-through was collected and remaining protein eluted three times with 5 mL TEV buffer followed by 7 mL TEV buffer supplemented with 0.1% CHAPS (Calbiochem 220201). Eluted fractions were combined, passed through a 0.22 µm syringe filter (Elkay E25-PS22-50S), concentrated to 0.5 mL using a 100 k MWCO concentrator (Millipore UFC810024), and injected onto an S200 10/300 GL gel filtration column equilibrated with 20 mM HEPES pH 8.0, 300 mM NaCl, 0.5 mM TCEP. Eluted fractions were inspected by SDS-PAGE, selected fractions were pooled, concentrated, frozen in liquid nitrogen in 2 µL aliquots, and stored at –80 °C until use. Mutant VPS34 complexes and complexes with an N-terminal mCherry tag on BECLIN1 were purified in the same way as wild type complexes.

## p40$^{phox}$ PX domain purification and labelling

To prepare the p40$^{phox}$ PX domain, plasmid pYO1645 was transformed into *E. coli* OverExpress C41(DE3) chemically competent cells (Lucigen 60442). Transformed bacteria were cultured in 6 L 2xTY medium supplemented with 0.1 mg/mL ampicillin (Formedium AMP50) at 37 °C until OD$_{600}$=0.8, then expression was induced with 0.5 mM IPTG, and bacteria were incubated for a further 19 hr at 18 °C. Cells were pelleted by centrifugation at 6700 × $g$ for 25 min and pellets were resuspended in 100 mL lysis buffer (50 mM HEPES pH 8.0, 150 mM NaCl, 12% glycerol, 10 mM DTT, 1 mM PMSF, 1 x EDTA-free protease inhibitor cocktail tablet). Cells were sonicated for 5 min on ice (2 s on, 3 s off, 60% amplitude), and lysate cleared by centrifugation at 142,000 g for 15 min at 4 °C in a Type 45 Ti rotor. Supernatant was filtered through a 5 µm syringe filter and incubated with 4 mL GS4B-GST beads (Cytiva 17–0756) equilibrated in lysis buffer for 1 hour at 4 °C at 8 rpm rotation. Beads were then transferred to a gravity flow column, washed with 50 mL lysis buffer, 200 mL wash buffer (50 mM HEPES pH 8.0, 300 mM NaCl, 12% glycerol, 10 mM DTT), and 50 mL TEV buffer (50 mM HEPES pH 8.0, 300 mM NaCl, 2 mM TCEP). Beads were resuspended in 10 mL TEV buffer, then 80 µL of TEV protease at 4.4 mg/mL was added, and samples were incubated overnight without rotation to cleave the N-terminal GST-tag. Flow-through and four 5 mL elutions with TEV buffer were pooled and supplemented with 20 mM imidazole, then filtered through a 0.22-µm syringe filter and incubated with 0.5 mL Ni-NTA agarose beads (Qiagen 1018244) equilibrated with TEV buffer for 45 minutes at 4 °C at 8 rpm rotation to remove the His$_6$-tagged TEV protease. Ni-NTA beads were transferred to a gravity flow column and the collected flow-through was again filtered through a 0.22-µm syringe filter and supplemented with 5 mM TCEP. Protein was concentrated to 1 mL using a 10 k MWCO concentrator (Millipore UFC901024) and injected onto an S75 16/60 gel filtration column equilibrated in 20 mM HEPES pH 8.0, 150 mM NaCl, 1 mM TCEP. The peak fractions were pooled and concentrated to 31.5 mg/mL.

To label the p40$^{phox}$ PX domain, 1 mg of AlexaFluor 647 C2 maleimide (Invitrogen A20347) was dissolved in 104 µL 100% DMSO (Fisher Bioreagents BP231) for a final concentration of 7.7 mM. Protein and dye were mixed in labelling buffer (20 mM HEPES pH 7.0, 200 mM KCl, 1 mM TCEP), with final concentration of p40$^{phox}$ PX domain at 112 µM and AF647 at 224 µM and incubated for 2 hr at room temperature (rt) with slow rotation. Reaction was quenched with 1 mM DTT and loaded onto a 5 mL HiTrap Heparin HP column (Cytiva 17-0406-01) equilibrated with 20 mM HEPES pH 7.4. The column was washed with 65 mL 20 mM HEPES pH 7.4 buffer and eluted with a gradient of Hep-B buffer (20 mM HEPES pH 7.4, 1 M KCl). Peak fractions were pooled and concentrated to 168 µM with 90% labelling efficiency.

## Human RAB1A purification and labelling

For maleimide labelling of human RAB1A, the pJB78 plasmid (GST-TEV-HsRAB1A Q70L C26S C126S, 1–204; *Tremel et al., 2021*) was used. To purify the protein, pJB78 was transformed into *E. coli* C41(DE3)RIPL cells, 2 L of transformed bacteria were cultured in 2xTY medium supplemented 0.1 mg/mL ampicillin at 37 °C until $OD_{600}$=0.6, then induced with 0.2 mM IPTG and grown for a further 18 hr at 18 °C. Cells were pelleted by centrifugation at $6700 \times g$ for 25 min, and pellets resuspended in 50 mL lysis buffer (25 mM HEPES pH 8.0, 200 mM NaCl, 1 mM TCEP, 0.5 mg/mL lysozyme (Sigma L6876), 0.05 μL/mL universal nuclease (Pierce 88702)). Cells were sonicated for 6 min (10 s on, 10 s off, 60% amplitude), and lysate cleared by centrifugation at $104,000 \times g$ for 45 min at 4 °C in a Type 45 Ti rotor (Beckmann Coulter). After filtering through a 0.45-μm syringe filter, lysate was incubated with 4 mL GS4B-GST beads equilibrated with lysis buffer for 1 hr at 4 °C at 8 rpm rotation. Beads were transferred to a gravity flow column and washed with 100 mL lysis buffer, then 100 mL wash buffer (20 mM HEPES pH 8.0, 300 mM NaCl, 5% glycerol, 1 mM TCEP). Beads were resuspended in 10 mL TEV buffer (25 mM HEPES pH 8.0, 150 mM NaCl, 1 mM TCEP), 60 μL of TEV protease at 4.4 mg/mL was added, and sample was incubated overnight at 4 °C without rotation. Flow-through and five 5 mL TEV buffer elutions were collected and diluted with Hep-A buffer (25 mM HEPES pH 8.0, 1 mM TCEP) to a final NaCl concentration of 50–100 mM, then applied to a 5 mL HiTrap Heparin HP column. The flow-through fraction with protein was concentrated and incubated with 16 x molar excess of GTP (Jena Bioscience NU-1012) or GDP (Sigma G7127) and 27 x molar excess EDTA for 90 minutes on ice, followed by addition of 55 x molar excess $MgCl_2$ and further incubation for 30 min on ice. Protein was applied onto an S75 16/60 gel filtration column equilibrated with 25 mM HEPES pH 7.0, 150 mM NaCl. Main peak was pooled and concentrated to 816 μM.

To label RAB1A for affinity binding assays, 1 mg of AlexaFluor 488 C5 maleimide (Invitrogen A10254) was dissolved in 138.8 μL 100% DMSO (Fisher Bioreagents BP231) for a final concentration of 10 mM. Protein and dye were mixed in labelling buffer (20 mM HEPES pH 7.0, 200 mM KCl, 1 mM TCEP), with final concentration of RAB1A at 112 μM and AF488 at 224 μM and incubated for 2 hr at rt with slow rotation. Reaction was quenched with 1 mM DTT and applied to an S75 16/60 gel filtration column equilibrated with 20 mM HEPES pH 7.0, 300 mM NaCl. Peak fractions were pooled and concentrated to 88 μM with 14% labelling efficiency.

## Human RAB5A purification

For maleimide labelling of human RAB5A, the pOP823 plasmid (HsRAB5A Q79L C19S,C63S, 1–212; *Tremel et al., 2021*) was used. To purify the protein, pOP823 was transformed into *E. coli* OverExpress C41(DE3) cells, 4 L of transformed bacteria were cultured in 2xTY medium supplemented with 0.1 mg/mL ampicillin at 37 °C until $OD_{600}$=0.6, then induced with 0.5 mM IPTG and grown for further 20 hr at 16 °C. Cells were pelleted by centrifugation at $6700 \times g$ for 25 min, and pellets were resuspended in 50 mL lysis buffer (25 mM HEPES pH 8.0, 200 mM NaCl, 10 mM imidazole, 1 mM TCEP, 0.5 mg/mL lysozyme, 0.05 μL/mL universal nuclease). Cells were sonicated for 7 min (10 s on, 10 s off, 60% amplitude), and lysate was cleared by centrifugation at $104,000 \times g$ for 45 min at 4 °C in a Type 45 Ti rotor (Beckmann Coulter). Supernatant was applied to a 5 mL HisTrap FF column (Cytiva 17-5255-01) equilibrated with Ni-A buffer (20 mM HEPES pH 8.0, 100 mM NaCl, 5% glycerol, 10 mM imidazole, 2 mM beta-mercaptoethanol (βME)). Protein was washed with 150 mL Ni-A buffer and eluted with a gradient of Ni-B buffer (20 mM HEPES pH 8.0, 100 mM NaCl, 5% glycerol, 200 mM imidazole, 2 mM βME). Eluted sample was dialysed in presence of ULP1 SUMO protease in dialysis buffer (20 mM HEPES pH 7.0, 150 mM NaCl, 5% glycerol, 1 mM TCEP) overnight at 4 °C in 10 k MWCO dialysis tubing (Thermo Fisher 68100). Dialysed protein was applied again on a 5 mL HisTrap FF column to remove the cleaved $His_6$-SUMO tag. Flow-through and seven 10 mL elutions with dialysis buffer were collected and concentrated using a 10 k MWCO concentrator. RAB5A was loaded with GTP or GDP, purified, and concentrated in the same way as RAB1A.

To label RAB5A for affinity binding assays, 1 mg of AlexaFluor 488 C5 maleimide (Invitrogen A10254) was dissolved in 138.8 μL 100% DMSO (Fisher Bioreagents BP231) for a final concentration of 10 mM. Protein and dye were mixed in labelling buffer (20 mM HEPES pH 7.0, 200 mM KCl, 1 mM TCEP), with final concentration of RAB5A at 112 μM and AF488 at 224 μM and incubated for 2 hr at rt with slow rotation. Reaction was quenched with 1 mM DTT and applied to an S75 16/60 gel filtration

column equilibrated with 20 mM HEPES pH 7.0, 300 mM NaCl. Final RAB5A-AF488 product was concentrated to 146 µM with 43% labelling efficiency.

## Purification of mCherry-tagged yeast VPS34-CII

Plasmids carrying genes encoding the subunits of the WT ScVPS34-CII (pYO359+pYO1922) or ScVPS34-CII carrying a RAB5-binding mutant (pYO1920+pYO1922) were transformed into the YOY193 yeast strain [*ADE2::TRX1* Bcy123 (*pep4::HIS3 prb1::LEU2 bar1::HISG lys2::Gal1/10-Gal4 can1 ade2 trp1 his3 ura3 leu2-3,112 lys2*); (*Rostislavleva et al., 2015*)]. Cells were grown in YM4 medium (0.67% yeast nitrogen base without amino acids; 0.5% casamino acids and 2% glucose, supplemented with 0.002% adenine sulfate; and 0.002% tyrosine). Gene expression was induced by 2% galactose in the presence of 2% glycerol, 3% lactic acid, and 10 mM $ZnCl_2$ without glucose at 30 °C for 26 hr. Cells were lysed in lysis buffer [50 mM Tris at pH 8.8, 150 mM NaCl, 0.5 mM TCEP, 1% Triton-X, 0.5 mM phenylmethylsulfonyl fluoride (PMSF), and a protease inhibitor tablet (Roche 05056489001)] with zirconium beads using a cell disruptor (FastPrep-24, MP Biomedicals) at the intensity of 6.5 for 40 s. Lysates were spun at $3000 \times g$ for 1 min. Supernatants were transferred to 2 ml tubes, and spun at $21,000 \times g$ for 5 min at 4 °C on a bench top centrifuge. Supernatants were combined and passed through a 5-mm syringe filter. Immunoglobulin G (IgG) beads (GE Healthcare 17096902) were added to the lysate, incubated for 3 hr at 4 °C at 8 rpm, then washed once with wash buffer (50 mM Tris at pH 8.0, 150 mM NaCl, 0.5 mM TCEP, 0.1% Triton-X, 50 mM $MgCl_2$, 5 mM ATP and 5 µg/mL RNaseA [Sigma, 83834]) and TEV buffer (50 mM Tris, pH 8.0, 150 mM NaCl, 0.5 mM TCEP). The protein A (ZZ) tags were cleaved by TEV protease in TEV buffer overnight. The elution fractions were concentrated and subjected to gel filtration on Superdex 200 10/30 equilibrated in 20 mM HEPES at pH 8.0, 300 mM NaCl, and 0.5 mM TCEP. The peak fractions were combined and concentrated using a 100 MWCO concentrator (Millipore UFC810024).

## Purification of Vps21 Q66L-EGFP

The pYO1919 plasmid was transformed into OverExpress C41(DE3) chemically competent cells (Lucigen 60442). Transformed bacteria were cultured in 4 L 2xTY medium supplemented with 0.1 mg/mL ampicillin (Formedium AMP50) at 37 °C until OD600=0.8, then expression was induced with 0.5 mM IPTG, and bacteria were incubated for a further 16 hr at 15 °C. Cells were pelleted by centrifugation at $6700 \times g$ for 15 min, frozen in liquid nitrogen, and stored at –80 °C until use. Pellets were resuspended in 50 mL lysis buffer (50 mM HEPES pH 8.0, 150 mM NaCl, 12% glycerol, 5 mM beta-mercaptoethanol (BME), 1 mM PMSF, 1 x EDTA-free protease inhibitor cocktail tablet). Cells were sonicated for 5 min on ice (2 s on, 3 s off, 60% amplitude), and lysate was cleared by centrifugation at $142,000 \times g$ for 15 min at 4 °C in a Type 45 Ti rotor. Supernatant was filtered through a 5-µm syringe filter then passed through a 5 ml Ni-NTA FF column (GE Healthcare). The column was washed with 50 mL wash buffer (50 mM Tris pH 8.0, 300 mM NaCl, 5 mM BME, and 12% glycerol), then the protein was eluted with 20 mL elution buffer (50 mM HEPES pH 8.5, 20 mM NaCl, 300 mM imidazole, and 5 mM BME). The eluted protein was immediately loaded on a 5 mL Q column (GE Healthcare). The bound protein was eluted with a NaCl gradient between Q-A buffer (50 mM Tris pH 8.5 and 5 mM BME) and Q-B buffer (50 mM Tris pH 8.5, 1 M NaCl, and 5 mM BME). Elution fractions were combined and concentrated down to 219 mg/mL (4.3 mM). The protein was diluted to 1 mM, and incubated with 11 mM GTP and 18 mM EDTA at rt for 90 min, then 36 mM MgCl2 was added and further incubated for 30 min at rt. The GTP-loaded protein was subjected to gel filtration on a S75 16/60 column in GF buffer (20 mM HEPES 8.0, 150 mM NaCl, and 0.5 mM TCEP). After combining the peak fractions, 10 mM TCEP at a final concentration was added and the protein was concentrated down to 130 mg/mL (2.57 mM).

## Multiple sequence alignment

Sequences for human RAB GTPases were retrieved from UniProtKB (https://www.uniprot.org/; [UniProt, 2025]) and multiple sequence alignments generated using Clustal Omega (https://www.ebi.ac.uk/jdispatcher/msa/clustalo *Sievers and Higgins, 2021*). Alignments were inspected and corrected using JalView (*Waterhouse et al., 2009*), and results were analysed and visualised with ESPript 3.0 (https://espript.ibcp.fr/ESPript/ESPript/; *Gouet et al., 1999*).

## GUV assay

GUVs were prepared using a hydrogel-assisted swelling protocol. 88 µL of 5% polyvinyl alcohol (PVA, Sigma 8.14894, MW approx. 145 kDa) solution was deposited onto a 13 mm cleaned coverslip (VWR 631–0150) and a thin layer of PVA was created by centrifuging for 30 s in a fixed-speed microfuge (IKA mini G, 6000 rpm) using a custom-made adaptor for cover slips. PVA-coated coverslips were dried at 60 °C for 30 min, and 15 µL of GUV lipid mixture at 1 mg/mL lipid concentration was deposited on the PVA layer. Coverslips were desiccated for 60 min, and GUVs swollen for 1 hr at rt in 220 µL 500 mM sterilised sucrose solution. Freshly generated GUVs were transferred to a 1.5 mL tube passivated by incubating with 1 mL of 5 mg/mL BSA (Sigma A7030) for 1 hr, followed by two 500 mM sucrose rinses.

To immobilise the GUVs, chambers of an eight-well glass bottom plate (Ibidi 80807) were incubated with 150 µL of avidin solution (0.1 mg/mL egg-white avidin, Invitrogen A2667, dissolved in PBS, and 1 mg/mL BSA) for 15 min at rt, and washed three times with 150 µL observation buffer (50 mM HEPES pH 7.0, 241.4 mM NaCl). 64 µL observation buffer was mixed with 48 µL GUVs in each chamber and incubated overnight at 4 °C in presence of 10 µM (fourfold molar excess to the PE-MCC lipid) RAB protein. GUVs were carefully washed four times with 360 µL wash buffer (31.8 mM HEPES pH 8.0, 172.7 mM NaCl, 181.8 mM sucrose, 5 mM BME).

GUVs were observed with a 63 x oil-immersion objective (Plan-Apochromat 63 x/1.4 Oil DIC M27, Zeiss) on an inverted confocal microscope (LSM 880 Indimo, AxioObserver, Zeiss), using ZEN software (Zeiss). The observation chamber was immobilised on a universal mounting frame (PeCon 010–800 486) using a chamber system insert plate (PeCon 000 470). After confirming that GUVs were immobilised on the wells, 20 µL of 10 x buffer was added (250 mM HEPES pH 8.0, 20 mM MnCl$_2$, 10 mM TCEP, 1 mM ATP pH 8.0, 10 mM EGTA in assays with EGTA). In ZEN software, Time Series and Positions were selected, and Definite Focus.2 (Zeiss) was used to maintain focus for every time point and position. The Lissamine-Rhodamine channel for GUV visualisation was excited using a diode-pumped solid-state (DPSS) 561 nm laser and collected with a 570–623 nm band. The AlexaFluor 647 channel for PX domain detection was excited using a HeNe 633 nm laser and collected with a 638–755 nm band. Four areas were selected per well to analyse at least 10 GUVs. To start the reaction, 48 µL of a solution containing the VPS34 kinase complex, labelled PX domain, in protein dilution buffer (25 mM HEPES pH 8.0, 150 mM NaCl, 1 mM TCEP, 0.67 mg/mL BSA) was added to the chamber. No BSA was added in the protein dilution buffer for assays without BSA. The final concentration of PX domain was 3.7 µM. The final concentrations of VPS34 complexes are described in figures and their legends. Images of each position were taken every 120 s for 2–3 hr. Images were analysed using the strategy and macro described in *Ohashi et al., 2020*.

## Confocal microscopy bead-based assay for RAB protein binding to VPS34 complexes

10–30 µL RFP-trap agarose beads (Chromotek RTA-10) were washed with 100 µL binding buffer (20 mM HEPES pH 8.0, 100 mM NaCl, 0.5 mM TCEP) and incubated with 2 µM mCherry-tagged VPS34 complex for 30 min on ice. Excess buffer was removed after centrifugation at 500 × *g* for 30 s and beads were washed with 100 µL binding buffer to remove unbound VPS34 complex. 30 µL reactions were prepared in PCR tubes with final bead concentration of 2% and AF488-labelled RAB proteins were added at varying final concentrations. Reactions were incubated on ice for 30 min and transferred to a 384-well plate (Greiner 781856) for imaging.

Beads were imaged with a 10 x objective (EC Plan-Neofluar 10 x/0.3 M27, Zeiss) on an inverted confocal microscope (LSM 880 Indimo, AxioObserver, Zeiss), using ZEN software (Zeiss). The 384-well plate was immobilised on a universal mounting frame (PeCon 010–800 486). DIC channel was used to visualise beads. The mCherry channel was excited using a diode-pumped solid-state (DPSS) 561 nm laser and collected with a 578–696 nm band. The AlexaFluor 488 channel was excited using an Argon 488 nm laser and collected with a 493–543 nm band. Images were analysed using Fiji software. Beads were selected using the ellipse ROI (region of interest) selection tool and added to the ROI manager, and fluorescence intensity of mCherry and AF488 channels was measured using a modified version of macro described by *Ohashi et al., 2020*. Background intensity was measured from 10 ROIs without beads and averaged to subtract from measured bead fluorescence intensities. For each bead, intensity of the AF488 channel was normalised by mCherry channel intensity to account for unequal mCherry-tagged VPS34 complex loading on the RFP-trap beads. Data were further analysed with

GraphPad Prism (GraphPad) and binding affinities estimated using the non-linear fitting tool with 'One site – specific binding' option. The reaction curves with error bars are presented in *Figure 1—figure supplement 4*, *Figure 2—figure supplement 1*, and *Figure 2—figure supplement 4*.

## Nano differential scanning fluorimetry

To determine the melting temperatures ($T_m$) of VPS34 complexes, nanoDSF experiments were carried out on the Prometheus NT.48 instrument (NanoTemper). 11 μL samples were prepared with final VPS34 complex concentration of 0.5 mg/mL in SEC buffer (20 mM HEPES pH 8.0, 0.5 mM TCEP, 150 mM NaCl for complex I or 300 mM NaCl for complex II) and incubated on ice for 30 min. Prometheus standard capillaries (NanoTemper PR-C002) were filled with 10 μL sample and placed on the sample holder. Prior to the experiment, fluorescence signal was optimised by adjusting the excitation power. A temperature gradient was then applied at 1 °C/min rate, from 15°C to 90°C, and intrinsic protein fluorescence emissions recorded at 330 nm and 350 nm. Data were visualised and analysed in GraphPad Prism (GraphPad).

## Yeast strains

The *S. cerevisiae* BY4741 (MATa *his3Δ1 leu2Δ0 met15Δ0 ura3Δ0*) strain was from Open Biosystems. The *vps15Δ* null yeast was created by a PCR-based gene targeting with the primers yoo1652 (5'-GTTGAATTGGAGAAATACAAAAGCTGTAAGGTTATCAAAAAGGAAGGCATACAGTATAgtttagcttg ctcgtccccgcc-3') and yoo1653 (5'-GAATACAAATTTATACGCATTTAGAATAAAGAAGCTGaaactggatgg cggcgttagtat-3'), and the *Schizosaccharomyces pombe HIS5* gene as a template (*Longtine et al., 1998*). For fluorescence microscopy, a *vps15Δ* strain integrated with a plasmid carrying N-terminally tandem-tomato-tagged VPS21 Q66L (pYO1879) was created by linearising the plasmid with NheI enzyme (YOY415).

## Yeast plasmid transformation

For each transformation, 5–10 absorbance units at 600 nm ($A_{600}$) of yeast cells either from plates or liquid culture were suspended in 100 μL One-step buffer (40% PEG3350, 0.2 M lithium acetate, 10 mM Tris pH 7.4, 100 mM DTT), and 3 μL of a plasmid was added to the mixture and incubated at rt for 10 min. The mixture was then incubated at 42 °C for 30 min, spun at 3000 × *g* for 30 s to remove the buffer. The yeast pellet was resuspended in 50 mL $H_2O$ and spread on a -URA plate.

## CPY assay

Yeast cells were grown to log phase ($A_{600}$=0.5–0.9) at 30 °C in -URA medium. Five $A_{600}$ units of cells were collected and resuspended in 500 μL of -URA medium containing 0.5 mg/mL bovine serum albumin (BSA) and 50 mM KPO4 at pH 5.7 and incubated at 30 °C for 1 hr. Cell and medium fractions were separated by centrifugation at 3000 × *g* for 1 min. Medium fractions were precipitated with final 10% trichloroacetic acid (TCA, from 100% stock) on ice for 15 min and centrifuged at 4 °C at 20,000 × *g* for 10 min. Pellets were washed once with 90% cold acetone. Both fractions were resuspended in 100 μL of lysis buffer (50 mM Tris at pH 8.0, 1% SDS, 8 M urea, 10 mM DTT, and 5 mM EDTA), 30 μL of 4 x sample buffer, and 0.5 mm zirconia beads (YTZ GRINDING MEDIA, Tosh). Cells and TCA pellets were disrupted at 5000 revolutions per minute (rpm) for 30 s with PowerLyzer (Mo Bio) at 4 °C. Samples were boiled at 98 °C for 2 min and loaded on SDS-PAGE gel.

CPY is synthesised as a premature form (p1), glycosylated in the Golgi (p2), and then delivered to the vacuole, where it is cleaved to become a mature form (m). The m and p2 forms of CPY were detected both in cellular and medium fractions using anti-CPY (1/1000; Invitrogen A6428). The C-terminally FLAG-tagged Vps15 proteins were detected by anti-Flag-HRP (HRP, horseradish peroxidase) (1/1000; Sigma A8592). Pgk1 was detected by anti-Pgk1 antibody (1/1000; Abcam ab113687). The bands were detected using ChemiDoc Touch Imaging System (Bio-Rad) with Chemiluminescence mode.

For each condition, intensities of p2 and m bands in cell and medium fractions were summed as a total intensity, then the percentages of m and p2 were calculated. In the case p2 was detected both in cell and medium fractions, the percentage of the summed intensity from these fractions was shown. The band intensities were measured using Image Lab 6.0 (Bio-Rad). The quantification was from three independent experiments.

## Yeast fluorescence microscopy

The YOY415 strain was transformed with the plasmids carrying WT VPS15-EGFP (pYO634) or RAB5-binding mutant VPS15-EGFP (pYO1880). Cells were grown to mid-log phase (A600=0.5–0.9), then examined with an inverted microscope (Nikon Eclipse Ti2) equipped with a CMOS camera (HAMA-MATSU ORCA-Flash4.0 C11440) and a niji LED Light Source (bluebox Optics). Images were analysed and levels were adjusted using Fiji.

## Hydrogen-deuterium exchange coupled to mass spectrometry

HDX-MS experiments were carried out as follows: Complex II (48 µM) alone or in the presence of 90 µM RAB5A-Q79L-C19S-C63S-GTP were pre-incubated for 30 min on ice. Complex II alone sample was then exposed for different deuteration times (3 sec on ice, 3, 30, 300, and 3000 s at rt) to $D_2O$ only buffer (20 mM Hepes pD8, 300 mM NaCl) to yield solution that was 5 µM Complex II and $D_2O$ at 83% final concentration. Complex II +RAB5A sample was exposed for the same deuteration times to a $D_2O$+RAB5A buffer (20 mM Hepes pD8, 300 mM NaCl, 90 µM RAB5A) to yield solution that was 5 µM Complex II +90 µM RAB5A and $D_2O$ at 83% final concentration. Samples were then immediately flash-frozen in liquid nitrogen and stored at –80 °C until analysis. Prior to LC-MS analysis, samples were quickly thawed and injected on a HDX Manager coupled to an Acquity UPLC M-Class system (Waters) set at 0.1 °C. Samples were then digested online at 15 °C using a 20 mm x 2.0 mm home-made pepsin column (Pepsin from Thermo Fisher Scientific, 20343 and hardware from Upchurch Scientific, C130-B) and loaded on a UPLC pre-column (ACQUITY UPLC BEH C18 VanGuard pre-column, 2.1 mm I.D. ×5 mm, 1.7 µm particle diameter, Waters, 186003975) for 2 min at 100 µL/min. Digested peptides were then eluted over 20 min from the pre-column onto an ACQUITY UPLC BEH C18 column (1.0 mm I.D. ×100 mm, 1.7 µm particle diameter, Waters, 186002346) using a 5–43% gradient of acetonitrile (Romil, H050), 0.1% formic acid. MS data were acquired using a Synapt G2Si-HDMS (Waters) with electrospray ionisation, using the data-independent acquisition mode (HDMS$^E$) over an *m/z* range of 50–2000 and a 100 fmol/µL Glu-FibrinoPeptide solution was used for lock-mass correction and calibration. The following parameters were used during the acquisition: capillary voltage, 3 kV; sampling cone voltage, 40 V; source temperature, 90 °C; desolvation gas, 150 °C and 650 L.h$^{-1}$; scan time, 0.3 s; trap collision energy ramp, 20–45 eV. Peptide identification was performed using ProteinLynx Global Server 3.0.1 (PLGS, Waters) with a home-made protein sequence library containing VPS34, VPS15, UVRAG, BECLIN1, RAB5A-Q79L-C19S-C63S-GTP, and pepsin sequences, with peptide and fragment tolerances set automatically by PLGS. Peptides were filtered using HDExaminer 3.4.0 (Trajan Scientific and Medical) with a minimum fragment per amino acid of 0.2, a minimum intensity of $10^3$, a length between 5 and 25 residues, a minimum PLGS score of 6.62, and an MH$^+$ tolerance of 5 ppm. An initial automated spectral processing step was conducted by HDExaminer followed by a manual inspection of individual peptides for sufficient quality where only one charge state per peptide was kept. Deuterium uptakes were not corrected for back-exchange and are reported as relative. HDX-MS results were statistically validated using an unpaired two-tailed t-test, where significant differences required 0.3 Da, 5% and p-value of <0.05 (n=3). A table with peptides within the dataset and the quality assessment statistics are available in *Figure 1—figure supplement 6—source data 1*. HDX-MS results were illustrated on a Complex II model using PyMOL 2.5.4 (https://www.pymol.org/). The mass spectrometry proteomics data (including raw data, table of every peptide included within the dataset and all relative uptake plots) have been deposited to the ProteomeXchange Consortium via the PRIDE partner repository (*Perez-Riverol et al., 2022*) with the dataset identifier PXD061277.

## Cryo-ET and subtomogram averaging

We re-processed the cryo-ET dataset of VPS34-CII-BATS on RAB5-coated LUVs described in *Tremel et al., 2021*. The tilt series raw images were imported in Relion 5.0 (beta4-commit-dfea29, *Burt et al., 2024*) – tomo mode, and motion corrected using Relion 5.0 implementation of MOTIONCOR2 (*Zheng et al., 2017*). CTF was estimated using the built-in CTFFIND-4.1 (*Rohou and Grigorieff, 2015*), and tilt series were aligned using the IMOD (*Kremer et al., 1996*) wrapper with fiducial based alignment. 10 Å/pixel tomograms were reconstructed for picking, and particles on membranes were manually picked on three representative tomograms. CrYOLO 1.9.9 (*Wagner et al., 2019*) was used for picking in tomograms after training on the manually picked particles using the trained custom model described above as pretrained weights, with a threshold of 0.05 and a box size of 220 Å.

146,464 particles were re-imported in Relion 5.0, scaled to the raw movies and extracted at bin 8 with a box size of 375 Å. The EMD-12237 density was used as a reference for 3D classification using a low-pass filter of 60 Å, the best 3D class was selected with 23,237 particles and subjected to another round of 3D classification using bin 4 particles. Two classes with 8441 particles were selected and refined at bin 1, masking out the membrane and using Blush regularisation. One refinement round of CTF refinement followed by Bayesian polishing was done and reached a resolution of 9.88 Å. To look for the RAB5 binding site in VPS15, a 3D classification was done on the 23,237 bin4 particles using the SPA structure of VPS34-CII with two RAB5 molecules a VPS34-CII on the membrane, a 3D classification was done on the same particles using the membrane bilayer as a reference.

## Cryo-EM sample preparation and screening

### VPS34-CII (VPS34 199-REIE-202 > 199-AAAA-202) + RAB5A and VPS34-CII (VPS34 199-REIE-202 > 199-ERIR-202) + RAB5A

VPS34-CII at the final concentration of 6 µM was incubated with 90 µM RAB5A for 20 min on ice in buffer containing 50 mM HEPES pH 8.0, 200 mM NaCl, 1 mM TCEP, 5 mM $MgCl_2$, and 1 mM AMP-PNP. The sample was crosslinked with BS3 at a final concentration of 30 µM for further 20 min on ice, then quenched with 100 mM TRIS pH 8.0. Prior to blotting, the sample was supplemented with 0.005% (w/w) Nonidet P-40 substitute and 4 mM CHAPSO. 3.5 µL sample was applied to a glow-discharged (40 mA, 90 s, Edwards Sputter Coater S150B) UltrAuFoil R 1.2/1.3 Au300 grid and plunge-frozen in liquid ethane using a Vitrobot Mk 3 (Thermo Fisher) plunger at 14 °C, 100% humidity, with 0 s waiting time, 4 s blotting time, and blot force set to +8. Grids were screened on a 200 kV Glacios (Thermo Fisher) microscope with a Falcon 3 detector in linear mode.

### WT VPS34-CII + RAB5A

VPS34-CII at the final concentration of 4 µM was incubated with 90 µM RAB5A for 20 min on ice in buffer containing 50 mM HEPES pH 8.0, 200 mM NaCl, 1 mM TCEP, 5 mM $MgCl_2$, and 1 mM AMP-PNP. The sample was crosslinked with BS3 at a final concentration of 20 µM for further 20 min on ice, then quenched with 100 mM TRIS pH 8.0. Prior to blotting, the sample was supplemented with 0.005% (w/w) Nonidet P-40 substitute and 4 mM CHAPSO. 3.5 µL sample was applied to a glow-discharged (40 mA, 90 s, Edwards Sputter Coater S150B) UltrAuFoil R 1.2/1.3 Au300 grid and plunge-frozen in liquid ethane using a Vitrobot Mk 3 (Thermo Fisher) plunger at 14 °C, 100% humidity, with 0 s waiting time, 4 s blotting time, and blot force set to +8. Grids were screened on a 200 kV Glacios (Thermo Fisher) microscope with a Falcon 3 detector in linear mode.

## Cryo-EM data collection

### VPS34-CII (VPS34 199-REIE-202 > 199-AAAA-202) + RAB5A

Images were collected on a 300 kV Titan Krios G3 TEM (Thermo Fisher) with a Gatan K3 detector (Gatan) in counting mode and energy filter set to 20 eV. 10,739 movies were collected at ×105,000 magnification and pixel size of 0.86 Å/px using EPU software (Thermo Fisher) in AFIS mode, with two movies collected per foil hole. The defocus range used was –0.8 nm to –2.2 nm, and the dose rate was 15.1 e⁻/px/s over an exposure length of 1.96 s, making the total dose 40 e⁻/Å. Movies were fractionated into 40 frames.

### VPS34-CII (VPS34 199-REIE-202 > 199-ERIR-202) + RAB5A

Images were collected on a 300 kV Titan Krios G3 TEM (Thermo Fisher) with a Gatan K3 detector (Gatan) in counting mode and energy filter set to 20 eV. 13,301 movies were collected at ×105,000 magnification and pixel size of 0.86 Å/px using EPU software (Thermo Fisher) in AFIS mode, with three movies collected per foil hole. The defocus range used was –0.8 nm to –2.2 nm, and the dose rate was 26.4 e⁻/px/s over an exposure length of 1.12 s, making the total dose 40 e⁻/Å. Movies were fractionated into 40 frames.

### WT VPS34-CII + RAB5A

Images were collected on a 300 kV Titan Krios G1 TEM (Thermo Fisher) with a Gatan K3 detector (Gatan) in counting mode and energy filter set to 20 eV. In total, 19,844 movies were collected on two

grids, at ×105,000 magnification and pixel size of 0.73 Å/px using EPU software (Thermo Fisher) in AFIS mode, with three movies collected per foil hole. The defocus range used was –0.8 nm to –2.2 nm. For the first grid, the dose rate was 15 e⁻/px/s over an exposure length of 1.42 s, making the total dose 40 e⁻/Å. For the second grid, the dose rate was 21.5 e⁻/px/s over an exposure length of 1 s, making the total dose 40 e⁻/Å. Movies were fractionated into 40 frames.

## Cryo-EM data processing
### VPS34-CII (VPS34 199-REIE-202 > 199-AAAA-202) + RAB5A (*Figure 1—figure supplement 3*)

Data was pre-processed using RELION 4.0. Movie frames were motion-corrected using MotionCorr2, and CTF of motion-corrected micrographs was estimated using CTFFIND-4.1. Micrographs with CTF-estimated maximum resolution of ≤5 Å were selected for further processing. Particle picking was done with CrYOLO-1.7.5 using a custom-trained model. Particles were extracted with twofold downsampling and at a box size of 204x204 px and imported to CryoSPARC for 2D classification. Selected particles were converted from .cs to star and re-imported to RELION for re-extraction at full box size of 408x408 px. Full size particles were again imported into CryoSPARC and used for homogeneous refinement with the density of VPS34-CI low-passed to 40 Å as an initial reference, yielding a 3.43 Å map after FSC-mask auto-tightening (EMD-54316). An initial model for VPS34-CII (REIE >AAAA)+RAB5A was built into this density by rigid-body fitting PDB depositions of VPS34 kinase domain (PDB: 3IHY) and RAB5A (PDB: 1R2Q), AlphaFold2 predictions of VPS15, VPS34 C2 domain, BECLIN1 CC2-BARA domains, UVRAG CC2-BARA2-CTD domains, and AlphaFold3 prediction of BECLIN1 CC1 domain and UVRAG-CC1 domain. Models were consolidated into a single entity for VPS34-CII (REIE >AAAA)+RAB5A using Coot 0.9.8.7. From this initial model, UCSF ChimeraX command 'molmap' was used to generate masks on the adaptor arm, kinase arm, base of the complex, and the RAB5A binding site on VPS34. The particles from the homogeneous refinement job in Cryo-SPARC were classified into eight 3D classes, and selected classes (see *Figure 1—figure supplement 3* for details) were used to refine regions of the complex mentioned above. The adaptor arm was refined to 3.2 Å resolution (EMD-54312); the kinase arm yielded a map with resolution of 3.15 Å (EMD-54313); the map of the base of the complex had a resolution of 3.2 Å (EMD-54314); the RAB5A binding site on VPS34 was refined to 3.25 Å (EMD-54315). Maps from the consensus and local refinements were processed using CryoTEN. Local refinement maps were merged into a composite map in UCSF ChimeraX (EMD-54317, EMD-54358). The initial model for VPS34-CII (REIE >AAAA)+RAB5A was refined using the composite map with ISOLDE, StarMap, TEMPy-ReFF, PHENIX, and Servalcat. Model was validated through MolProbity and deposited to PDB as 9*R*X5.

Two of the eight classes (classes 1 and 6) from the final 3D classification job were later found to contain RAB5A bound to VPS15 subunit. These densities were used as initial references for heterogeneous refinement in the processing of VPS34-CII (REIE >ERIR)+RAB5A and wild-type VPS34-CII+RAB5A datasets. One class did not contain RAB5A bound to the complex, and this density was used as an 'apo VPS34-CII' reference.

### VPS34-CII (VPS34 199-REIE-202 > 199-ERIR-202) + RAB5A (*Figure 1—figure supplement 5*)

Movie frames were motion-corrected in RELION 4.0 with MotionCorr2, and CTF of motion-corrected micrographs was estimated using CTFFIND-4.1. Micrographs with CTF-estimated maximum resolution of ≤5 Å were selected for further processing. Particle picking was done with CrYOLO-1.7.5 using a custom-trained model. Picked particles were extracted with twofold downsampling and a box size of 220x220 px, then imported to CryoSPARC for heterogeneous refinement with five initial references: a noise decoy, apo VPS34-CII, VPS34-CII+RAB5A on VPS34, VPS34-CII+RAB5A on VPS15, and VPS34-CII with two RAB5A bound simultaneously. The particles sorted into the classes with noise decoy and VPS34-CII + (RAB5A)₂ volumes did not produce meaningful densities and were removed. To find particles that had RAB5A bound to VPS34, the other three classes were subject to 3D classification into two classes with a focus mask on the RAB5A binding site on VPS34, and 'force hard classification' parameter set to 'true'. The class with a density resembling a RAB5A bound to VPS34 was poorly resolved and further processing of the particles associated with this density did not yield a meaningful map. Therefore, the class without a density for RAB5A on VPS34 was selected for further

processing. These particles were again force classified into two 3D classes, with a focus mask on RAB5A bound to VPS15. Particles from the class with an extra density on the VPS15 subunit were re-imported to RELION and re-extracted at full size with a box size of 440x440 px, then imported back to CryoSPARC for non-uniform refinement that yielded a map at 3.68 Å after the FSC-mask auto-tightening (EMD-54321). An initial model was built into this density by fitting the model of VPS34-CII (REIE >AAAA)+RAB5A. The RAB5A chain was moved from the binding site on VPS34 subunit to the VPS15 subunit and fitted into the density. This model was used in UCSF ChimeraX to generate masks for local refinements via the 'molmap' command. The kinase arm was refined to 3.55 Å (EMD-54318); the adaptor arm to 4.09 Å (EMD-54319), and the local refinement of the RAB5A binding site on the VPS15 subunit yielded a map at 3.52 Å (EMD-54320). Maps from the consensus and local refinements were processed using CryoTEN. Densities from local refinements were combined into a composite map on UCSF ChimeraX (EMD-54322, EMD-54359), which was used to refine the initial model for VPS34-CII (REIE >ERIR)+RAB5A with ISOLDE, StarMap, TEMPy-ReFF, PHENIX, and Servalcat. The model was validated through MolProbity and deposited to PDB as 9*RX*6.

## WT VPS34-CII + RAB5A (*Figure 1—figure supplement 7*)

Two datasets were collected for this sample. For both datasets, the motion-correction was done in RELION 4.0 with MotionCorr2, and the CTF of motion-corrected micrographs was estimated using CTFFIND-4.1. Micrographs with CTF-estimated maximum resolution of ≤5 Å were selected for further processing. Particle picking was done with CrYOLO-1.7.5 using a custom-trained model. Picked particles were extracted with two-fold downsampling and a box size of 220x220 px, then imported to CryoSPARC.

For dataset 1, two rounds of heterogeneous refinement were performed with five initial references: a noise decoy, apo VPS34-CII, VPS34-CII+RAB5A on VPS34, VPS34-CII+RAB5A on VPS15, and VPS34-CII with two RAB5A bound simultaneously. Least noisy classes were further classified into ten 3D classes. Particles from seven classes were re-imported into RELION and re-extracted at full size with a box size of 440x440 px, then imported back to CryoSPARC for merging with dataset 2.

For dataset 2, picked particles were subject to 2D classification, and then used in homogeneous refinement together with VPS34-CI as the initial reference. Particles were then classified into two 3D classes with a focus mask on the RAB5A binding site on VPS34. The class with a more prominent density on the VPS34 RAB5 binding site was used in two rounds of heterogeneous refinement, with four initial references: VPS34-CII+RAB5A on VPS15, VPS34-CII+RAB5A on VPS34, VPS34-CII with two RAB5A bound simultaneously, and apo VPS34-CII. Particles from the best densities were re-imported to RELION, re-extracted at full size with the box size of 440x440 px, imported back to CryoSPARC and merged with particles from dataset 1.

Combined 233,866 particles were subject to two rounds of heterogeneous refinement using five initial references: a noise decoy, apo VPS34-CII, VPS34-CII+RAB5A on VPS34, VPS34-CII+RAB5A on VPS15, and VPS34-CII with two RAB5A bound simultaneously. 66,323 particles were classed as apo VPS34-CII, with no density for RAB5A visible on VPS34 or VPS15 at any viewing threshold level on UCSF ChimeraX. These particles were used in non-uniform refinement and yielded a 3.87 Å reconstruction of apo VPS34-CII (EMD-54361). 49,688 particles were assigned to the class with RAB5A bound to the VPS34 subunit. To confirm that in this particle set (A) there were no particles with RAB5A bound to VPS15 and (B) all particles had RAB5A bound to the VPS34 site, several rounds of 'hard' 3D classifications were performed with two classes and focus masks on the RAB5A binding site either on VPS34 or VPS15 subunits. No extra density on VPS15 was present, and none of the particles had an empty VPS34 RAB5A binding site; therefore, all particles were used in a non-uniform refinement job that yielded a 3.99 Å map of VPS34-CII with RAB5A bound to VPS34 (EMD-54362).

44,127 particles contained VPS34-CII with the extra RAB5A density on the VPS15 subunit. These particles were further inspected using the procedure described above to ensure that (A) no particles had RAB5A bound to the VPS34 subunit, and (B) all particles had RAB5A bound to VPS15 subunit. This particle set was merged with the 26,318 particles found during the processing of the class with VPS34-CII + (RAB5A)$_2$ described below. The total of 70,445 particles for the density of VPS34-CII+RAB5A on the VPS15 subunit were used in non-uniform refinement that yielded a reconstruction at 4.00 Å (EMD-54363), 58,253 particles that were classed as VPS34-CII with two RAB5A GTPases were used in a 'hard' 3D classification into two classes with a focus mask on the VPS34 RAB5A binding

site. The class that did not have a RAB5A density on the VPS34 subunit contained 26,318 particles that were merged with the other 44,127 particles for VPS34-CII+RAB5A on VPS15 density. The other class with 31,935 particles was subject to further inspection as described above to confirm that all particles in this set (A) had RAB5A bound to VPS15 and (B) had RAB5A bound to the VPS34 site. All particles were confirmed to have two RAB5A proteins bound to VPS34-CII and were used for a non-uniform refinement job that produced a map at 4.03 Å resolution (EMD-54364).

Maps from the consensus refinements were processed using CryoTEN. Models for VPS34-CII apo, VPS34-CII+RAB5A on VPS34, VPS34-CII+RAB5A on VPS15, and VPS34-CII+RAB5A on VPS34 and VPS15 were built by fitting a model of VPS34-CII (REIE >AAAA)+RAB5A into the density and adjusting the RAB5A chain appropriately. Models were refined using ISOLDE, StarMap, TEMPy-ReFF, PHENIX, and Servalcat. Models were validated through MolProbity and deposited to PDB as 9*RX*8 (apo VPS34-CII), 9*RX*9 (VPS34-CII+RAB5A on VPS34), 9RXA (VPS34-CII+RAB5A on VPS15), and 9RXB (VPS34-CII+RAB5A on VPS34 and VPS15). Consistent with what was reported previously for VPS34-CI (*Cook et al., 2025*), our highest resolution cryo-EM structures contain a density continuous with the VPS15 N-terminus that is consistent with a myristate moiety. When this density was not apparent, the reconstructions had lower resolution. We also observed density for a GDP nucleotide in the VPS15 kinase domain. As it has been reported that VPS15 binds a GTP nucleotide in its active site (*Cook et al., 2025*), it may be that the GTP had been hydrolysed, and we were seeing the bound product of the reaction (*Figure 4—figure supplement 2B*).

### PDB models

Cryo-EM maps and refined VPS34-CII models were deposited in EMDB and PDB under accession codes:

WT VPS34-CII: PDB 9*RX*8, EMD-54361;
WT VPS34-CII-RAB5A$_{VPS34}$: PDB 9*RX*9, EMD-54362;
WT VPS34-CII-RAB5A$_{VPS15}$: PDB 9RXA, EMD-54363;
WT VPS34-CII-RAB5A$_{VPS34,VPS15}$: PDB 9RXB, EMD-54364;
VPS34-CII REIE >AAAA-RAB5A$_{VPS34,VPS15}$: PDB 9*RX*5, EMD-54358, EMD-54317, EMD-54316, EMD-54313, EMD-54312, EMD-54314, EMD-54315;
VPS34-CII REIE >ERIR-RAB5A$_{VPS15}$: PDB 9*RX*6, EMD-54359, EMD-54322, EMD-54321, EMD-54318, EMD-54319, EMD-54320;

Reprocessed tomography map for VPS34-CII(ΔBATS)-RAB5A$_{VPS34}$ bound to membrane (*Tremel et al., 2021*): EMD-54360, and reinterpreted model: PDB 9S47.

## Source code availability

The macro script used for GUV assays is available from the below link: https://github.com/MRC-LMB-Light-Microscopy-Facility/guv-macro/ (copy archived at *Boulanger, 2025*).

## Acknowledgements

We thank Anna Yeates, Grigory Sharov, Bilal Ahsan, Giuseppe Cannone of the MRC LMB EM facility, and Madhanagopal Anandapadamanaban for help with cryo-EM data collection, Jake Grimmett, Toby Darling and Ivan Clayson for assistance and advice with scientific computing, Stephen McLaughlin and Chris Batters for help with biophysical instrumentation and experiment design, Olga Perisic for helpful discussions, Jerome Boulanger for the modified microscope macros, the Light Microscopy Facility for help with confocal microscopy, the mechanical workshop for custom instruments, the BioMass Facility for help with MS instruments, and Garib Murshudov for assistance with model refinement. The work was supported by the Medical Research Council (MC_U105184308 to RLW) and Cancer Research UK (grant DRCPGM\100014 to RLW).

## Additional information

### Funding

| Funder | Grant reference number | Author |
|---|---|---|
| Medical Research Council Laboratory of Molecular Biology | MC_U105184308 | Roger L Williams |
| Cancer Research UK | DRCPGM\100014 | Roger L Williams |

The funders had no role in study design, data collection and interpretation, or the decision to submit the work for publication.

### Author contributions

Saule Spokaite, Conceptualization, Formal analysis, Validation, Investigation, Visualization, Methodology, Writing – original draft, Writing – review and editing; Yohei Ohashi, Conceptualization, Formal analysis, Validation, Investigation, Visualization, Methodology, Writing – review and editing; Maxime Bourguet, Formal analysis, Validation, Investigation, Visualization, Methodology, Writing – original draft, Writing – review and editing; Antoine Nicolas Dessus, Validation, Investigation, Visualization, Methodology, Writing – review and editing; Roger L Williams, Conceptualization, Supervision, Funding acquisition, Validation, Investigation, Visualization, Methodology, Writing – original draft, Project administration, Writing – review and editing

### Author ORCIDs

Saule Spokaite ⓘ https://orcid.org/0000-0001-9461-7957
Yohei Ohashi ⓘ https://orcid.org/0000-0002-2288-130X
Maxime Bourguet ⓘ https://orcid.org/0000-0002-3614-0553
Antoine Nicolas Dessus ⓘ https://orcid.org/0009-0003-2517-0188
Roger L Williams ⓘ https://orcid.org/0000-0001-7754-4207

Reviewer #1 (Public review): https://doi.org/10.7554/eLife.110040.3.sa1
Reviewer #2 (Public review): https://doi.org/10.7554/eLife.110040.3.sa2
Reviewer #3 (Public review): https://doi.org/10.7554/eLife.110040.3.sa3
Author response https://doi.org/10.7554/eLife.110040.3.sa4

---

## Additional files

### Supplementary files

MDAR checklist

Supplementary file 1. Supplementary data.

### Data availability

Cryo-EM maps and refined VPS34-CII models were deposited in EMDB and PDB under accession codes: WT VPS34-CII: PDB 9RX8, EMD-54361; WT VPS34-CII-RAB5AVPS34: PDB 9RX9, EMD-54362; WT VPS34-CII-RAB5AVPS15: PDB 9RXA, EMD-54363; WT VPS34-CII-RAB5AVPS34,VPS15: PDB 9RXB, EMD-54364; VPS34-CII REIE>AAAA-RAB5AVPS34,VPS15: PDB 9RX5, EMD-54358, EMD-54317, EMD-54316, EMD-54313, EMD-54312, EMD-54314, EMD-54315; VPS34-CII REIE>ERIR-RAB5AVPS15: PDB 9RX6, EMD-54359, EMD-54322, EMD-54321, EMD-54318, EMD-54319, EMD-54320; Reprocessed tomography map for VPS34-CII(ΔBATS)-RAB5AVPS34 bound to membrane (*Tremel et al., 2021*): EMD-54365.

The following datasets were generated:

| Author(s) | Year | Dataset title | Dataset URL | Database and Identifier |
|---|---|---|---|---|
| Bourguet M | 2024 | A novel RAB5 binding site in human VPS34-CII that is likely the primordial site in eukaryotic evolution | https://www.ebi.ac.uk/pride/archive/projects/PXD061277 | PRIDE, PXD061277 |
| Spokaite S, Ohashi Y, Dessus AN, Bourguet M, Williams RL | 2025 | VPS34-CII (VPS34 199-REIE-202 to 199-AAAA-202 mutant) bound to RAB5A (Q79L) | https://doi.org/10.2210/pdb9RX5/pdb | Worldwide Protein Data Bank, 10.2210/pdb9RX5/pdb |
| Spokaite S, Ohashi Y, Dessus AN, Bourguet M, Williams RL | 2025 | VPS34-CII (VPS34 199-REIE-202 to 199-ERIR-202 mutant) bound to RAB5A (Q79L) on the VPS15 subunit | https://doi.org/10.2210/pdb9RX6/pdb | Worldwide Protein Data Bank, 10.2210/pdb9RX6/pdb |
| Spokaite S, Ohashi Y, Dessus AN, Bourguet M, Williams RL | 2025 | Apo VPS34-CII (VPS34/VPS15/BECLIN1/UVRAG) | https://doi.org/10.2210/pdb9RX8/pdb | Worldwide Protein Data Bank, 10.2210/pdb9RX8/pdb |
| Spokaite S, Ohashi Y, Dessus AN, Bourguet M, Williams RL | 2025 | VPS34-CII bound to RAB5A-GTP 1-212 (C19S, C63S, Q79L) on the VPS34 subunit | https://doi.org/10.2210/pdb9RX9/pdb | Worldwide Protein Data Bank, 10.2210/pdb9RX9/pdb |
| Spokaite S, Ohashi Y, Dessus AN, Bourguet M, Williams RL | 2025 | VPS34-CII bound to RAB5A-GTP 1-212 (C19S, C63S, Q79L) on the VPS15 subunit | https://doi.org/10.2210/pdb9RXA/pdb | Worldwide Protein Data Bank, 10.2210/pdb9RXA/pdb |
| Spokaite S, Ohashi Y, Dessus AN, Bourguet M, Williams RL | 2025 | VPS34-CII (VPS34/VPS15/BECLIN1/UVRAG) bound to RAB5A (Q79L) on the VPS34 and VPS15 subunits | https://doi.org/10.2210/pdb9RXB/pdb | Worldwide Protein Data Bank, 10.2210/pdb9RXB/pdb |
| Tremel S, Dessus AN, Williams RL | 2025 | Human complex II-BATS bound to membrane-attached Rab5a-GTP | https://doi.org/10.2210/pdb9S47/pdb | Worldwide Protein Data Bank, 10.2210/pdb9S47/pdb |
| Spokaite S, Ohashi Y, Dessus AN, Bourguet M, Williams RL | 2025 | Apo VPS34-CII (VPS34/VPS15/BECLIN1/UVRAG), consensus refinement map processed with CryoTEN | https://www.ebi.ac.uk/emdb/EMD-54361 | Electron Microscopy Data Bank, EMD-54361 |
| Spokaite S, Ohashi Y, Dessus AN, Bourguet M, Williams RL | 2025 | VPS34-CII bound to RAB5A-GTP 1-212 (C19S, C63S, Q79L) on the VPS34 subunit, primary map processed with CryoTEN | https://www.ebi.ac.uk/emdb/EMD-54362 | Electron Microscopy Data Bank, EMD-54362 |
| Spokaite S, Ohashi Y, Dessus AN, Bourguet M, Williams RL | 2025 | VPS34-CII bound to RAB5A-GTP 1-212 (C19S, C63S, Q79L) on the VPS15 subunit, primary map processed with CryoTEN | https://www.ebi.ac.uk/emdb/EMD-54363 | Electron Microscopy Data Bank, EMD-54363 |
| Spokaite S, Ohashi Y, Dessus AN, Bourguet M, Williams RL | 2025 | VPS34-CII (VPS34/VPS15/BECLIN1/UVRAG) bound to RAB5A (Q79L) on the VPS34 and VPS15 subunits, primary map processed with CryoTEN | https://www.ebi.ac.uk/emdb/EMD-54364 | Electron Microscopy Data Bank, EMD-54364 |
| Spokaite S, Ohashi Y, Dessus AN, Bourguet M, Williams RL | 2025 | VPS34-CII (VPS34 199-REIE-202 to 199-AAAA-202 mutant) bound to RAB5A (Q79L), primary composite map processed with CryoTEN | https://www.ebi.ac.uk/emdb/EMD-54358 | Electron Microscopy Data Bank, EMD-54358 |

*Continued*

| Author(s) | Year | Dataset title | Dataset URL | Database and Identifier |
|---|---|---|---|---|
| Spokaite S, Ohashi Y, Dessus AN, Bourguet M, Williams RL | 2025 | VPS34-CII (VPS34 199-REIE-202 > 199-AAAA-202 mutant) bound to RAB5A-GTP (Q79L); unsharpened composite map | https://www.ebi.ac.uk/emdb/EMD-54317 | Electron Microscopy Data Bank, EMD-54317 |
| Spokaite S, Ohashi Y, Dessus AN, Bourguet M, Williams RL | 2025 | VPS34-CII (VPS34 199-REIE-202 > 199-AAAA-202 mutant) bound to RAB5A-GTP (Q79L), consensus refinement | https://www.ebi.ac.uk/emdb/EMD-54316 | Electron Microscopy Data Bank, EMD-54316 |
| Spokaite S, Ohashi Y, Dessus AN, Bourguet M, Williams RL | 2025 | VPS34-CII (VPS34 199-REIE-202 to 199-AAAA-202 mutant) bound to RAB5A (Q79L, focused refinement on the kinase arm) | https://www.ebi.ac.uk/emdb/EMD-54313 | Electron Microscopy Data Bank, EMD-54313 |
| Spokaite S, Ohashi Y, Dessus AN, Bourguet M, Williams RL | 2025 | VPS34-CII (VPS34 199-REIE-202 to 199-AAAA-202 mutant) bound to RAB5A (Q79L, focused refinement on the adaptor arm) | https://www.ebi.ac.uk/emdb/EMD-54312 | Electron Microscopy Data Bank, EMD-54312 |
| Spokaite S, Ohashi Y, Dessus AN, Bourguet M, Williams RL | 2025 | VPS34-CII (VPS34 199-REIE-202 to 199-AAAA-202 mutant) bound to RAB5A (Q79L), focused refinement on the base of the complex | https://www.ebi.ac.uk/emdb/EMD-54314 | Electron Microscopy Data Bank, EMD-54314 |
| Spokaite S, Ohashi Y, Dessus AN, Bourguet M, Williams RL | 2025 | VPS34-CII (VPS34 199-REIE-202 to 199-AAAA-202 mutant) bound to RAB5A (Q79L, focused refinement on the interface between RAB5A and VPS34) | https://www.ebi.ac.uk/emdb/EMD-54315 | Electron Microscopy Data Bank, EMD-54315 |
| Spokaite S, Ohashi Y, Dessus AN, Bourguet M, Williams RL | 2025 | VPS34-CII (VPS34 199-REIE-202 to 199-ERIR-202 mutant) bound to RAB5A (Q79L) on the VPS15 subunit, composite map processed with CryoTEN | https://www.ebi.ac.uk/emdb/EMD-54359 | Electron Microscopy Data Bank, EMD-54359 |
| Spokaite S, Ohashi Y, Dessus AN, Bourguet M, Williams RL | 2025 | VPS34-CII (VPS34 199-REIE-202 to 199-ERIR-202 mutant) bound to RAB5A (Q79L) on the VPS15 subunit; unsharpened composite map | https://www.ebi.ac.uk/emdb/EMD-54322 | Electron Microscopy Data Bank, EMD-54322 |
| Spokaite S, Ohashi Y, Dessus AN, Bourguet M, Williams RL | 2025 | VPS34-CII (VPS34 199-REIE-202 to 199-ERIR-202 mutant) bound to RAB5A (Q79L) on the VPS15 subunit, consensus refinement | https://www.ebi.ac.uk/emdb/EMD-54321 | Electron Microscopy Data Bank, EMD-54321 |

*Continued on next page*

*Continued*

| Author(s) | Year | Dataset title | Dataset URL | Database and Identifier |
|---|---|---|---|---|
| Spokaite S, Ohashi Y, Dessus AN, Bourguet M, Williams RL | 2025 | VPS34-CII (VPS34 199-REIE-202 to 199-ERIR-202 mutant) bound to RAB5A (Q79L) on the VPS15 subunit, focused refinement on the kinase arm | https://www.ebi.ac.uk/emdb/EMD-54318 | Electron Microscopy Data Bank, EMD-54318 |
| Spokaite S, Ohashi Y, Dessus AN, Bourguet M, Williams RL | 2025 | VPS34-CII (VPS34 199-REIE-202 to 199-ERIR-202 mutant) bound to RAB5A (Q79L) on the VPS15 subunit, focused refinement on the adaptor arm | https://www.ebi.ac.uk/emdb/EMD-54319 | Electron Microscopy Data Bank, EMD-54319 |
| Spokaite S, Ohashi Y, Dessus AN, Bourguet M, Williams RL | 2025 | VPS34-CII (VPS34 199-REIE-202 to 199-ERIR-202 mutant) bound to RAB5A (Q79L) on the VPS15 subunit, focused refinement of the RAB5A interface with VPS15 | https://www.ebi.ac.uk/emdb/EMD-54320 | Electron Microscopy Data Bank, EMD-54320 |
| Dessus AN, Tremel S, Williams RL | 2025 | Human complex II-BATS bound to membrane-attached Rab5a-GTP | https://www.ebi.ac.uk/emdb/EMD-54560 | Electron Microscopy Data Bank, EMD-54560 |
| Dessus AN, Tremel S, Williams RL | 2025 | VPS34-CII-BATS bound to membrane-attached RAB5A-GTP, re-processed data | https://www.ebi.ac.uk/emdb/EMD-54365 | Electron Microscopy Data Bank, EMD-54365 |

The following previously published datasets were used:

| Author(s) | Year | Dataset title | Dataset URL | Database and Identifier |
|---|---|---|---|---|
| Tremel S, Morado DR, Kovtun O, Williams RL, Briggs JAG, Munro S, Ohashi Y, Bertram J, Perisic O | 2021 | human complex II-BATS bound to membrane-attached Rab5a-GTP | https://doi.org/10.2210/pdb7BL1/pdb | Worldwide Protein Data Bank, 10.2210/pdb7BL1/pdb |
| Tremel S, Ohashi Y, Morado DR, Bertram J, Perisic O, Brandt LTL, von Wrisberg M-K, Chen ZA, Maslen SL, Kovtun O, Skehel M, Rappsilber J, Lang K, Munro S, Briggs JAG, Williams RL | 2021 | Subtomogram average reconstruction of humanVPS34 complex II bound to Rab5a on a lipid membrane | https://www.ebi.ac.uk/emdb/EMD-12237 | Electron Microscopy Data Bank, EMD-12237 |
| Tremel S, Morado DR | 2021 | human complex II-BATS bound to membrane-attached Rab5a-GTP | https://www.ebi.ac.uk/emdb/EMD-12214 | Electron microscopy data bank, EMD-12214 |

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

# Appendix 1

**Appendix 1—table 1.** Cryo-EM data collection, refinement and validation statistics for VPS34-CII with a 199-REIE-202 > AAAA mutation in the VPS34 subunit, bound to RAB5A.

| | VPS34-CII AAAA +RAB5A (EMD-54358) (PDB 9RX5) | VPS34-CII AAAA +RAB5A Adaptor arm (EMD-54312) | VPS34-CII AAAA +RAB5A Kinase arm (EMD-54313) | VPS34-CII AAAA +RAB5A Base of the complex (EMD-54314) | VPS34-CII AAAA +RAB5A RAB binding site (EMD-54315) |
|---|---|---|---|---|---|
| Data collection and processing | | | | | |
| Magnification | 105,000 | 105,000 | 105,000 | 105,000 | 105,000 |
| Voltage (kV) | 300 | 300 | 300 | 300 | 300 |
| Electron exposure (e–/Å2) | 40 | 40 | 40 | 40 | 40 |
| Defocus range (μm) | –0.8 to –2.2 | –0.8 to –2.2 | –0.8 to –2.2 | –0.8 to –2.2 | –0.8 to –2.2 |
| Pixel size (Å) | 0.86 | 0.86 | 0.86 | 0.86 | 0.86 |
| Symmetry imposed | C1 | C1 | C1 | C1 | C1 |
| Initial particle images (no.) | 492,761 | 492,761 | 492,761 | 492,761 | 492,761 |
| Final particle images (no.) | 203,301 | 136,898 | 151,160 | 88,897 | 88,897 |
| Map resolution (Å) FSC threshold | 3.2 0.143 | 3.2 0.143 | 3.15 0.143 | 3.25 0.143 | 3.25 0.143 |
| Map resolution range (Å) | 3.2–10.3 | 2.8–14.0 | 3.15–16.1 | 3.3–6.6 | 3.3–6.6 |
| | | | | | |
| Refinement | | | | | |
| Initial model used (PDB code) | 7BL1 | | | | |
| Model resolution (Å) FSC threshold | 3.6 0.5 | | | | |
| Map sharpening B factor (Å2) | –127.3 | –112.5 | –107.8 | –109.0 | –109.9 |
| Model composition | | | | | |
| Non-hydrogen atoms | 25436 | | | | |
| Protein residues | 3150 | | | | |
| Ligands | 7 | | | | |
| B factors (Å2) | | | | | |
| Protein | 88.6 | | | | |
| Ligand | 82.2 | | | | |

*Appendix 1—table 1 Continued on next page*

*Appendix 1—table 1 Continued*

| | VPS34-CII AAAA +RAB5A (EMD-54358) (PDB 9RX5) | VPS34-CII AAAA +RAB5A Adaptor arm (EMD-54312) | VPS34-CII AAAA +RAB5A Kinase arm (EMD-54313) | VPS34-CII AAAA +RAB5A Base of the complex (EMD-54314) | VPS34-CII AAAA +RAB5A RAB binding site (EMD-54315) |
|---|---|---|---|---|---|
| R.m.s. deviations | | | | | |
| Bond lengths (Å) | 0.0112 | | | | |
| Bond angles (°) | 0.78 | | | | |
| Validation | | | | | |
| MolProbity score | 1.23 | | | | |
| Clashscore | 1.99 | | | | |
| Poor rotamers (%) | 0.04 | | | | |
| Ramachandran plot | | | | | |
| Favored (%) | 96.2 | | | | |
| Allowed (%) | 3.7 | | | | |
| Disallowed (%) | 0.13 | | | | |

**Appendix 1—table 2.** Cryo-EM data collection, refinement and validation statistics for VPS34-CII with a 199-REIE-202 > ERIR mutation in the VPS34 subunit, bound to RAB5A.

| | VPS34-CII ERIR +RAB5A (EMD-54359) (PDB 9RX6) | VPS34-CII ERIR +RAB5A RAB binding site (EMD-54320) | VPS34-CII ERIR +RAB5A Kinase arm (EMD-54318) | VPS34-CII ERIR +RAB5A Adaptor arm (EMD-54319) |
|---|---|---|---|---|
| Data collection and processing | | | | |
| Magnification | 105,000 | 105,000 | 105,000 | 105,000 |
| Voltage (kV) | 300 | 300 | 300 | 300 |
| Electron exposure (e–/Å$^2$) | 40 | 40 | 40 | 40 |
| Defocus range (µm) | –0.8 to –2.2 | –0.8 to –2.2 | –0.8 to –2.2 | –0.8 to –2.2 |
| Pixel size (Å) | 0.86 | 0.86 | 0.86 | 0.86 |
| Symmetry imposed | C1 | C1 | C1 | C1 |
| Initial particle images (no.) | 370,889 | 370,889 | 370,889 | 370,889 |
| Final particle images (no.) | 84,296 | 84,296 | 84,296 | 84,296 |
| Map resolution (Å) FSC threshold | 3.7 0.143 | 3.5 0.143 | 3.55 0.143 | 4.1 0.143 |
| Map resolution range (Å) | 3.1–13.6 | 3.0–15.5 | 3.0–11.7 | 3.2–17.5 |
| Refinement | | | | |
| Initial model used (PDB code) | VPS34-CII AAAA (9RX5) | | | |
| Model resolution (Å) FSC threshold | 4.0 0.5 | | | |
| Map sharpening B factor (Å$^2$) | -114.4 | -111.3 | -113.6 | -134.0 |

*Appendix 1—table 2 Continued on next page*

*Appendix 1—table 2 Continued*

|  | VPS34-CII ERIR +RAB5A (EMD-54359) (PDB 9RX6) | VPS34-CII ERIR +RAB5A RAB binding site (EMD-54320) | VPS34-CII ERIR +RAB5A Kinase arm (EMD-54318) | VPS34-CII ERIR +RAB5A Adaptor arm (EMD-54319) |
|---|---|---|---|---|
| Model composition |  |  |  |  |
| Non-hydrogen atoms | 24072 |  |  |  |
| Protein residues | 2977 |  |  |  |
| Ligands | 5 |  |  |  |
| B factors (Å²) |  |  |  |  |
| Protein | 189 |  |  |  |
| Ligand | 115 |  |  |  |
| R.m.s. deviations |  |  |  |  |
| Bond lengths (Å) | 0.0112 |  |  |  |
| Bond angles (°) | 0.83 |  |  |  |
| Validation |  |  |  |  |
| MolProbity score | 1.34 |  |  |  |
| Clashscore | 2.54 |  |  |  |
| Poor rotamers (%) | - |  |  |  |
| Ramachandran plot |  |  |  |  |
| Favored (%) | 95.6 |  |  |  |
| Allowed (%) | 4.2 |  |  |  |
| Disallowed (%) | 0.17 |  |  |  |

**Appendix 1—table 3.** Cryo-EM data collection, refinement and validation statistics for wild type VPS34-CII apo form, bound to RAB5A on VPS34, bound to RAB5A on VPS15, and bound to RAB5A on both VPS34 and VPS15.

|  | Apo VPS34-CII (EMD-54361) (PDB 9RX8) | VPS34-CII+RAB5A (VPS34 site) (EMD-54362) (PDB 9RX9) | VPS34-CII+RAB5A (VPS15 site) (EMD-54363) (PDB 9RXA) | VPS34-CII+RAB5A (VPS34 and VPS15 sites) (EMD-54364) (PDB 9RXB) |
|---|---|---|---|---|
| Data collection and processing |  |  |  |  |
| Magnification | 105,000 | 105,000 | 105,000 | 105,000 |
| Voltage (kV) | 300 | 300 | 300 | 300 |
| Electron exposure (e–/Å²) | 40 | 40 | 40 | 40 |
| Defocus range (μm) | –0.8 to –2.2 | –0.8 to –2.2 | –0.8 to –2.2 | –0.8 to –2.2 |
| Pixel size (Å) | 0.73 | 0.73 | 0.73 | 0.73 |
| Symmetry imposed | C1 | C1 | C1 | C1 |
| Initial particle images (no.) | 717,560 | 717,560 | 717,560 | 717,560 |
| Final particle images (no.) | 66,323 | 49,688 | 70,445 | 31,935 |
| Map resolution (Å) FSC threshold | 3.87 0.143 | 3.99 0.143 | 4.00 0.143 | 4.03 0.143 |
| Map resolution range (Å) |  |  |  |  |
| Refinement |  |  |  |  |
| Initial model used (PDB code) | VPS34-CII AAAA (9RX5) | VPS34-CII AAAA (9RX5) | VPS34-CII AAAA (9RX5) | VPS34-CII AAAA (9RX5) |

*Appendix 1—table 3 Continued on next page*

*Appendix 1—table 3 Continued*

|  | Apo VPS34-CII (EMD-54361) (PDB 9RX8) | VPS34-CII+RAB5A (VPS34 site) (EMD-54362) (PDB 9RX9) | VPS34-CII+RAB5A (VPS15 site) (EMD-54363) (PDB 9RXA) | VPS34-CII+RAB5A (VPS34 and VPS15 sites) (EMD-54364) (PDB 9RXB) |
|---|---|---|---|---|
| Model resolution (Å) | 4.2 | 4.3 | 4.3 | 4.4 |
| FSC threshold | 0.5 | 0.5 | 0.5 | 0.5 |
| Map sharpening B factor (Å$^2$) | −131.3 | −134.3 | −134.7 | −118.7 |
| Model composition |  |  |  |  |
| Non-hydrogen atoms | 22714 | 24082 | 24070 | 25436 |
| Protein residues | 2809 | 2979 | 2977 | 3147 |
| Ligands | 3 | 5 | 5 | 7 |
| B factors (Å$^2$) |  |  |  |  |
| Protein | 200 | 199 | 216 | 205 |
| Ligand | 149 | 171 | 156 | 159 |
| R.m.s. deviations |  |  |  |  |
| Bond lengths (Å) | 0.0113 | 0.0160 | 0.0105 | 0.0103 |
| Bond angles (°) | 0.73 | 0.77 | 0.77 | 0.71 |
| Validation |  |  |  |  |
| MolProbity score | 1.16 | 1.15 | 1.23 | 1.23 |
| Clashscore | 1.83 | 1.79 | 2.2 | 2.36 |
| Poor rotamers (%) | 0.16 | 0.00 | 0.11 | 0.00 |
| Ramachandran plot |  |  |  |  |
| Favored (%) | 96.7 | 96.6 | 96.4 | 96.9 |
| Allowed (%) | 3.2 | 3.3 | 3.5 | 2.9 |
| Disallowed (%) | 0.14 | 0.14 | 0.11 | 0.19 |

**Appendix 1—table 4.** Yeast strains used in this study.

| ID | Genotype | Source |
|---|---|---|
| BY4741 | MATa his3Δ1 leu2Δ0 met15Δ0 ura3Δ0 | Open Biosystems |
| YOY412 | BY4741 Δvps15::HIS5 | This study |
| YOY415 | BY4741 Δvps15::HIS5 PHO5-td-tomato-vps21 (Q66L)-LEU2::VPS21 | This study |

