## [Editor Report · eLife Assessment]

This **convincing** study examines a novel interaction of RAB5 with VPS34 complex II. Structural data are combined with site-directed mutagenesis, sequence analysis, biochemistry, yeast mutant analysis, and prior data on RAB1-VPS34 and RAB5-VPS34 interactions to provide a new perspective on how RAB GTPases recruit related but distinct VPS34 complexes to different organelles. The judgment is that this work represents a **fundamental** advance in our understanding of VPS34 localization and regulation.

---

## [Referee Report · Reviewer #1 (Public review)]

Summary:

This manuscript presents high resolution cryoEM structures of VPS34-complex II bound to Rab5A at 3.2A resolution. The Williams group previously reported the structure of VPS34 complex II bound to Rab5A on liposomes using tomography, and therefore the previous structure, although very informative, was at lower resolution.

The first new structure they present is of the 'REIE>AAAA' mutant complex bound to RAB5A. The structure resembles the previously determined one except an additional molecule of RAB5A was observed bound to the complex in a new position, interacting with the solenoid of VPS15.

Although this second binding site exhibited reduced occupancy of RAB5A in the structure, the authors determined an additional structure in which the primary binding site was mutated to prevent RAB5A binding ('REIE>ERIR'). In this structure, there is no RAB5A bound to the primary binding site on VPS34, but the RAB5A bound to VPS15 now has strong density. The authors note that the way in which RAB5A interacts with each site is distinct, though both interfaces involve the switch regions. The authors confirm the location of this additional binding site using HDX-MS.

The authors then determine multiple structures of the wild-type complex bound to RAB5A from a single sample, as they use 3D classifications to separate out versions of the complex bound to 0, 1, or 2 copies of RAB5A. Overall the structure of VPS34-Complex II does not change between the different states, and the data indicate that both RAB5A binding sites can be occupied at the same time.

The authors then design a new mutant form of the complex (SHMIT>DDMIE) that is expected to disrupt the interaction at the secondary site between VPS15 and RAB5A. This mutation had a minor impact on the Kd for RAB5A binding, but when combined with the REIE>ERIR mutation of the primary binding site, RAB5A binding to the complex was abolished.

Comparison of sequences across species indicated that the RAB5A binding site on VPS15 was conserved in yeast while the RAB5A binding site on VPS34 is not.

The authors tested the impact of a correspond yeast Vps15 mutation (SHLITY>DDLIEY) predicted to disrupt interaction with yeast Rab5/Vps21, and found this mutant Vps15 protein was mislocalized and caused defective CPY processing.

The authors then compare these structures of the RAB5A-class II complex to recently published structures from the Hurley group of the RAB1A-class I complex, and find that in both complexes the Rab protein is bound to the VPS34 binding site in a somewhat similar manner. However, a key difference is the position of VPS34 is slightly different in the two complexes because of the unique ATL14L and UVRAG subunits in the class I and class II complexes, respectively. This difference creates a different RAB binding pocket that explains the difference in RAB specificity between the two complexes.

Finally, the higher resolution structures enable the authors to now model portions of BECLIN1 and UVRAG that were not previously modeled in the cryoET structure.

Strengths:

Overall I found this to be an interesting and comprehensive study of the structural basis for interaction of RAB5A with VPS34-complex II. The authors have performed experiments to validate their structural interpretations, and they present a clear and thorough comparative analysis of the Rab binding sites in the two different VPS34 complexes. The result is a much better understanding of how two different Rab GTPases specifically recruit two different, but highly similar complexes to the membrane surface.

Weaknesses:

No significant weaknesses noted.

---

## [Referee Report · Reviewer #2 (Public review)]

The work by Spokaite et al describes the discovery of a novel Rab5 binding site present in complex II of class III PI3K using a combination of HDX and Cryo EM. Extensive mutational and sequence analysis define this as the primordial Rab5 interface. The data presented are convincing that this is indeed a biologically relevant interface, and is important in defining mechanistically how vps34 complexes are regulated.

This paper is a very nice expansion of their previous cryo-ET work from 2021, and is an excellent companion piece on high resolution cryo-EM of the complex I class III complex bound to Rab1 from the Hurley lab in 2025. Overall, this work is of excellent technical quality, and answers important unexplained observations on some unexpected mutational analysis from the previous work.

They used their increased affinity vps34 mutant to determine the 3.2 ang structure of Rab5 bound to vps34-CII. Clear density was seen for the original Rab5 interface, but an additional site was observed. Based on this structure they mutated out the vps34 interface, allowing for a high resolution structure of the Rab5 bound at the Vps15 interface.

They extensively validated the vps15 interface in the yeast variant of vps34, showing that the Vp215-Rab5 (Vps21) interface identified is critical in controlling complex II vps34 recruitment.

The major strengths of this paper are that the experiments appear to be done carefully and rigorously and I have very few experimental suggestions.

Here is what I recommend based on some very minor weaknesses I observed

(1) My main concern has to do a little bit with presentation. My main issue is how the authors use mutant description. They clearly indicate the mutant sequence in the human isoform (for example see Fig 2A, Vps15 described as 579-SHMIT-583>DDMIE), however, when they shift to the yeast version they shift to saying vps15 mutant, but don't define the mutant, (Fig 2G). I would recommend they just include the same sequence numbering and WT to mutant replacement every time a new mutant (or species) is described. It is always easier to interpret what is being shown when the authors are jumping between species when the exact mutant is included. This is particularly important in this paper, where we are jumping between both different subunits and different species, so clear description in figure/figure legends makes it much easier to read for non-specialists.

(2) The HDX data very clearly shows that Rab5 is likely able to bind at both sites, which back ups the cryo EM data nicely. I am slightly confused by some of the HDX statements described in the methods.

(3) The authors state "Only statistically significant peptides showing a difference greater than 0.25 Da and greater than 5% for at least two timepoints were kept." This seems to be confusing why they required multiple timepoints, and before they also describe that they required a p value of less than 0.05. It might be clearer to state that significant differences required a 0.25 Da, 5%, and p value of <0.05 (n=3). Also what do they mean by kept? Does this mean that they only fully processed the peptides with differences.

(4) They show peptide traces for a selection in the supplement, but it would be ideal to include the full set of HDX data as an excel file, including peptides with no differences as there is a lot of additional information (deuteration levels for everything) that would be useful to share, as recommended from the Masson et al 2019 recommendations paper. This may be attached but this reviewer could not see an example of it in the shared data dropbox folder.

Comments on revisions:

The authors have addressed all of my issues.

---

## [Referee Report · Reviewer #3 (Public review)]

Summary:

The manuscript of Spokaite et al. focuses on the Vps34 complex involved in PI3P production. This complex exists in two variants, one (class I) specific for autophagy, and a second one (class II) specific for the endocytic system. Both differ only in one subunit. The authors previously showed that the Vps34 complexes interact with Rab GTPases, Rab1 or Rab5 (for class II), and the identified site was found at Vps34. Now, the authors identify a conserved and overlooked Rab5 binding site in Vps15, which is required for the function of the Class II complex. In support of this, they show cryo-EM data with a second Rab5 bound to Vps15, identify the corresponding residues, and show by mutant analysis that impaired Rab5 binding also results in defects using yeast as a model system.

Overall, this is a most complete study with little to criticize. The paper shows convincingly that the two Rab5 binding sites are required for Vps34 complex II function, with the Vps15 binding site being critical for endosomal localization. The structural data is very much complete. What I am missing are a few controls that show that the mutations in Vps15 do not affect autophagy. I also found the last paragraph of the results section a bit out of place, even though this is a nice observation that the N-terminal part of BECLIN has these domains. However, what does it add to the story?

Comments on revisions:

The authors answered all my questions. I have no further requests.

---

## [Author Response]

The following is the authors’ response to the original reviews.

**Public Reviews:**

**Reviewer #1 (Public review):**
Summary:This manuscript presents high-resolution cryoEM structures of VPS34-complex II bound to Rab5A at 3.2A resolution. The Williams group previously reported the structure of VPS34 complex II bound to Rab5A on liposomes using tomography, and therefore, the previous structure, although very informative, was at lower resolution.The first new structure they present is of the 'REIE>AAAA' mutant complex bound to RAB5A. The structure resembles the previously determined one, except that an additional molecule of RAB5A was observed bound to the complex in a new position, interacting with the solenoid of VPS15.Although this second binding site exhibited reduced occupancy of RAB5A in the structure, the authors determined an additional structure in which the primary binding site was mutated to prevent RAB5A binding ('REIE>ERIR'). In this structure, there is no RAB5A bound to the primary binding site on VPS34, but the RAB5A bound to VPS15 now has strong density. The authors note that the way in which RAB5A interacts with each site is distinct, though both interfaces involve the switch regions. The authors confirm the location of this additional binding site using HDX-MS.The authors then determine multiple structures of the wild-type complex bound to RAB5A from a single sample, as they use 3D classifications to separate out versions of the complex bound to 0, 1, or 2 copies of RAB5A. Overall, the structure of VPS34-Complex II does not change between the different states, and the data indicate that both RAB5A binding sites can be occupied at the same time.The authors then design a new mutant form of the complex (SHMIT>DDMIE) that is expected to disrupt the interaction at the secondary site between VPS15 and RAB5A. This mutation had a minor impact on the Kd for RAB5A binding, but when combined with the REIE>ERIR mutation of the primary binding site, RAB5A binding to the complex was abolished.Comparison of sequences across species indicated that the RAB5A binding site on VPS15 was conserved in yeast,while the RAB5A binding site on VPS34 is not.The authors tested the impact of a corresponding yeast Vps15 mutation (SHLITY>DDLIEY) predicted to disrupt interaction with yeast Rab5/Vps21, and found that this mutant Vps15 protein was mislocalized and caused defective CPY processing.The authors then compare these structures of the RAB5A-class II complex to recently published structures from the Hurley group of the RAB1A-class I complex, and find that in both complexes the Rab protein is bound to the VPS34 binding site in a somewhat similar manner. However, a key difference is that the position of VPS34 is slightly different in the two complexes because of the unique ATL14L and UVRAG subunits in the class I and class II complexes, respectively. This difference creates a different RAB binding pocket that explains the difference in RAB specificity between the two complexes.Finally, the higher resolution structures enable the authors to now model portions of BECLIN1 and UVRAG that were not previously modeled in the cryoET structure.Strengths:Overall, I found this to be an interesting and comprehensive study of the structural basis for the interaction of RAB5A with VPS34-complex II. The authors have performed experiments to validate their structural interpretations, and they present a clear and thorough comparative analysis of the Rab binding sites in the two different VPS34 complexes. The result is a much better understanding of how two different Rab GTPases specifically recruit two different, but highly similar complexes to the membrane surface.Weaknesses:No significant weaknesses were noted.
**Reviewer #2 (Public review):**
Summary:The work by Spokaite et al describes the discovery of a novel Rab5 binding site present in complex II of class III PI3K using a combination of HDX and Cryo EM. Extensive mutational and sequence analysis define this as the primordial Rab5 interface. The data presented are convincing that this is indeed a biologically relevant interface, and is important in defining mechanistically how VPS34 complexes are regulated.This paper is a very nice expansion of their previous cryo-ET work from 2021, and is an excellent companion piece on high-resolution cryo-EM of the complex I class III complex bound to Rab1 from the Hurley lab in 2025. Overall, this work is of excellent technical quality and answers important unexplained observations on some unexpected mutational analysis from the previous work.They used their increased affinity VPS34 mutant to determine the 3.2 ang structure of Rab5 bound to VPS34-CII. Clear density was seen for the original Rab5 interface, but an additional site was observed. Based on this structure, they mutated out the VPS34 interface, allowing for a high-resolution structure of the Rab5 bound at the VPS15 interface.They extensively validated the VPS15 interface in the yeast variant of VPS34, showing that the Vp215-Rab5 (VPS21) interface identified is critical in controlling complex II VPS34 recruitment.The major strengths of this paper are that the experiments appear to be done carefully and rigorously, and I have very few experimental suggestions.Here is what I recommend based on some very minor weaknesses I observed(1) My main concern has to do a little bit with presentation. My main issue is how the authors use mutant description. They clearly indicate the mutant sequence in the human isoform (for example, see Figure 2A, VPS15 described as 579-SHMIT-583>DDMIE); however, when they shift to the yeast version, they shift to saying VPS15 mutant, but don't define the mutant, (Figure 2G). I would recommend they just include the same sequence numbering and WT to mutant replacement every time a new mutant (or species) is described. It is always easier to interpret what is being shown when the authors are jumping between species, when the exact mutant is included. This is particularly important in this paper, where we are jumping between different subunits and different species, so a clear description in the figure/figure legends makes it much easier to read for non-specialists.

The reviewer has made an excellent point here. To clarify the yeast mutation, we have revised the manuscript main text to refer to the yeast mutant as SHLITY>DDLIEY, and we have added this to the legend for Figs. 2F,G.

(2) The HDX data very clearly shows that Rab5 is likely able to bind at both sites, which back ups the cryo EM data nicely. I am slightly confused by some of the HDX statements described in the methods.(3) The authors state, "Only statistically significant peptides showing a difference greater than 0.25 Da and greater than 5% for at least two timepoints were kept." This seems to be confusing as to why they required multiple timepoints, and before they also describe that they required a p-value of less than 0.05. It might be clearer to state that significant differences required a 0.25 Da, 5%, and p-value of <0.05 (n=3). Also, what do they mean by kept? Does this mean that they only fully processed the peptides with differences?(4) They show peptide traces for a selection in the supplement, but it would be ideal to include the full set of HDX data as an Excel file, including peptides with no differences, as there is a lot of additional information (deuteration levels for everything) that would be useful to share, as recommended from the Masson et al 2019 recommendations paper. This may be attached, but this reviewer could not see an example of it in the shared data dropbox folder.

We have revised the HDX method description to clarify. All peptides were kept and fully processed. However, for the results displayed, we have illustrated only peptides meeting the criteria described.

The Excel file for all peptides (as recommended by Masson et al) was deposited with PRIDE, with the identifier with the dataset identifier PXD061277, in addition, we have included this excel file in our supplementary material.

**Reviewer #3 (Public review):**
Summary:The manuscript of Spokaite et al. focuses on the Vps34 complex involved in PI3P production. This complex exists in two variants, one (class I) specific for autophagy, and a second one (class II) specific for the endocytic system. Both differ only in one subunit. The authors previously showed that the Vps34 complexes interact with Rab GTPases, Rab1 or Rab5 (for class II), and the identified site was found at Vps34. Now, the authors identify a conserved and overlooked Rab5 binding site in Vps15, which is required for the function of the Class II complex. In support of this, they show cryo-EM data with a second Rab5 bound to Vps15, identify the corresponding residues, and show by mutant analysis that impaired Rab5 binding also results in defects using yeast as a model system.Overall, this is a most complete study with little to criticize. The paper shows convincingly that the two Rab5 binding sites are required for Vps34 complex II function, with the Vps15 binding site being critical for endosomal localization. The structural data is very much complete.Weaknesses:What I am missing are a few controls that show that the mutations in Vps15 do not affect autophagy. I am wondering if this mutant is still functional in autophagy. This can be simply tested by sorting of Atg8 to the vacuole lumen using established assays or by following PhoΔ60 sorting. This analysis would reveal that the corresponding mutant is specific for the Class II complex.

One of the first noted features of the VPS34 complexes was that the ATG14-containing complex (VPS34-CI) is important for autophagy, while the VPS38 (yeast orthologue of UVRAG) subunit characteristic of VPS34-CII is important for endocytic sorting (PMID 11157979). However, the VPS34, VPS15 and BECLIN1 subunits are required are present in both complexes, as such, mutations of them may affect both processes.

We agree with the reviewer that is an important undertaking to examine the effect of the SHLITY>DDLIEY mutation in yeast Vps15 on autophagy. However, the focus of the current manuscript is VPS34-complex II and RAB5 interaction/activation. An autophagy effect would be more relevant for VPS34 complex I and RAB1. We have not presented any results for human VPS34-complex I - RAB1 nor yeast Vps34-complex I – Ypt1 (yeast RAB1 orthologue). We are preparing another manuscript focusing entirely on this, and it is not a simple story. While we think this is an important question, we believe that this is beyond the scope of the current manuscript.

It would be helpful if the authors could clarify whether they believe that Vps34 kinase activity is stimulated by Rab binding or whether this stimulation is a consequence of better membrane localization of Vps34. In other words, is the complex active with soluble PI3P in solution, and does the activity change if Rab5 is added to the complex? This might have been addressed in the past, but I did not see evidence for this, as the authors only addressed the activity of the Vps34 complexes on membranes.

The reviewer has raised an excellent question, which was addressed briefly in the introduction to the manuscript. We have now somewhat expanded on these issues near the end of the discussion in the revised manuscript. In our previously published study, we found that soluble RAB5-GTP did not stimulate the complex II activity (supplementary figure 2b of PMID: 33692360). This is consistent with our finding in this manuscript showing that RAB5 did not cause large conformational changes in solution. However, our previous single-molecule study showed that once complex II is recruited to the membrane by RAB5, and RAB5 increases the turnover rate on membranes, indicating an additional allosteric activation (Figure 7 of PMID: 33137306). This study indicated that the primary the role of RAB5 is to anchor complex II on the membrane. Once the complex is anchored on the membrane by RAB5, the kinase domain is in the vicinity of its substrate, PI, leading to higher turnover.

The Echelon Class III PI3K ELISA Kit (Echelon, K-3000) comes with a soluble PI, diC8 to measure the VPS34 activity, and it is certainly active with this soluble substrate. However, if the substrate is in membranes, the VPS34 activity is greatly dependent on the character of the membrane.

I also found the last paragraph of the results section a bit out of place, even though this is a nice observation that the N-terminal part of BECLIN has these domains. However, what does it add to the story?

The reviewer is correct that the high-resolution features of BECLIN1 at the base of the V-shaped complex that we observed are not related to RAB5 binding, but they are characteristic of VPS34-CII and likely to be important for the specific role of VPS34-CII. This is the first high-resolution structure of the VPS34-CII that has been reported, and we believe it would be irresponsible not to briefly describe them, since they are unique to VPS34-CII. For this reason, we have placed this section at the end of the results, and we now clarify that we do not see a relevance to RAB5 function, but we describe the arrangement of a region (the BH3) that has been functionally noted in many previous studies, in the absence of a structure.

**Reviewing Editor Comments:**
Please address the following suggestions for minor changes to the manuscript. Use your best scientific judgment in addressing the comments and describe the modifications together with your reasoning in a cover letter. We look forward to seeing the revised version of this very nice study.
**Recommendations for the authors:**

**Reviewer #1 (Recommendations for the authors):**
I found a portion of the description of the cryoEM complexes on the top of page 9 to be redundant with similar descriptions near the top of page 7, and it was not clear to me at first that these were describing the same structures. Part of my confusion was due to the redundancy, including the statement near the bottom of page 7: 'Models were built and refined for all RAB5associated VPS34-CII assemblies', and then the similar statement on page 9: 'We fit and refined atomic models into both densities'. I believe these are describing the same models? To clarify for the reader, perhaps on page 9, the authors could begin this part with a statement such as "as described above", and eliminate the redundant descriptions.

The reviewer is correct. Both sections describe the same set of cryo-EM classes from the same sample. The only difference is what we analysed in the two sections: number of RAB5s bound in the first section and the effect of RAB5 binding in the second section. We have revised the text to make this clear, and to make the second section more succinct.

**Reviewer #3 (Recommendations for the authors):**
(1) The authors show nicely that a mutation in Vps15 disrupts binding to Vps21 in vivo, with defects in the endocytic pathway as analyzed by CPY sorting. I am wondering if this mutant is still functional in autophagy. This can be simply tested by sorting of Atg8 to the vacuole lumen using established assays or by following Pho∆60 sorting. This analysis would reveal that the corresponding mutant is specific for the Class II complex. If the authors were to find evidence that this Vps15 mutant also affects autophagy, it would indicate that there is possibly also another Rab1 binding site in Vps15.

As we stated above, an autophagy effect would be more relevant for VPS34 complex I and RAB1. We have not presented any results for human VPS34-complex I - RAB1 nor yeast Vps34-complex I – Ypt1 (yeast RAB1 orthologue). We are preparing another manuscript focusing entirely on this, and it is not a simple story. While we think this is an important question, we believe that this is beyond the scope of the current manuscript.

(2) It would be helpful if the authors could clarify whether they believe that Vps34 kinase activity is stimulated by Rab binding or whether this stimulation is a consequence of better membrane localization of Vps34. In other words, is the complex active with soluble PI3P in solution, and does the activity change if Rab5 is added to the complex? This might have been addressed in the past, but I did not see evidence for this, as the authors only addressed the activity of the Vps34 complexes on membranes.

As in our response to reviewer #3 above, this point was addressed in previous publications and was described in the introduction to our manuscript.